



Solid Earth

# Estimating the depth and evolution of intrusions at resurgent calderas: Los Humeros (Mexico)

**Stefano Urbani[1], Guido Giordano[1,2], Federico Lucci[1], Federico Rossetti[1], Valerio Acocella[1], and Gerardo Carrasco-Núñez[3]**

[1]Dipartimento di Scienze, Università degli Studi Roma Tre, L.go S.L. Murialdo 1, 00146 Rome, Italy
[2]CNR – IDPA c/o Università degli Studi di Milano, Via Luigi Mangiagalli, 34, 20133 Milano, Italy
[3]Centro de Geociencias, Universidad Nacional Autónoma de México, Campus UNAM Juriquilla, 76100, Queretaro, Mexico

**Correspondence:** Stefano Urbani (stefano.urbani@uniroma3.it)

**Abstract.** [TS1]Resurgent calderas are excellent targets for geothermal exploration, as they are associated with the shallow emplacement of magma, resulting in widespread and long-lasting hydrothermal activity. Resurgence is classically attributed to the uplift of a block or dome resulting from the inflation of the collapse-forming magma chamber due to the intrusion of new magma. The Los Humeros volcanic complex (LHVC; Mexico) consists of two nested calderas: the outer and older Los Humeros formed at 164 ka and the inner Los Potreros formed at 69 ka. The latter is resurgent and currently the site of an active and exploited geothermal field (63 MWe installed). Here we aim to better define the characteristics of the resurgence in Los Potreros by integrating fieldwork with analogue models and evaluating the spatio-temporal evolution of the deformation as well as the depth and extent of the intrusions responsible for the resurgence, which may also represent the local heat source(s).

Structural field analysis and geological mapping show that the floor of the Los Potreros caldera is characterized by several lava domes and cryptodomes (with normal faulting at the top) that suggest multiple deformation sources localized in narrow areas.

Analogue experiments are used to define the possible source of intrusion responsible for the observed surface deformation. We apply a tested relationship between the surface deformation structures and depth of elliptical sources to our experiments with sub-circular sources. We found that this relationship is independent of the source and surface dome eccentricity, and we suggest that the magmatic sources inducing the deformation in Los Potreros are located at very shallow depths (hundreds of metres), which is in agreement with the well data and field observations. We propose that the recent deformation at LHVC is not a classical resurgence associated with the bulk inflation of a deep magma reservoir; rather, it is related to the ascent of multiple magma bodies at shallow crustal conditions (< 1 km depth). A similar multiple source model of the subsurface structure has also been proposed for other calderas with an active geothermal system (Usu volcano, Japan), suggesting that the model proposed may have wider applicability.

## 1 Introduction

Caldera resurgence consists of the post-collapse uplift of part of the caldera floor. Resurgence has been described in several calderas worldwide (Smith and Bailey, 1968; Elston, 1984; Lipman, 1984, and references therein), representing a frequent step in caldera evolution. Several mechanisms that trigger resurgence have been invoked, including the pressurization of the hydrothermal system (Moretti et al., 2018), regional earthquakes (Walter et al., 2009) and magmatic intrusion (Kennedy et al., 2012). Discriminating the contributions to the observed uplift of each of these mechanisms is often challenging (Acocella, 2014). However, despite the possible hydrothermal and tectonic contributions, field observations in eroded resurgent calderas (e.g. Tomochic, Swanson and McDowell, 1985; Kutcharo, Goto and McPhie, 2018; Turkey Creek, Du Bray and Pallister, 1999) coupled with the long timescale of the uplift of the caldera floor (from tens to thou-

sands of years) suggest that the intrusion of magmatic bodies is the prevalent mechanism for resurgence.

Resurgence is commonly attributed to the emplacement of silicic magmas at different depth levels under limited viscosity contrasts with regard to the previously emplaced magma (Marsh, 1984; Galetto et al., 2017). However, though rare, resurgence may also be triggered by the injection of more primitive magma (Morán-Zenteno et al., 2004; Kennedy et al., 2012) or by the emplacement of basaltic sills, as recently documented at the Alcedo caldera (Galapagos; Galetto et al., 2019). The shape of the intracaldera resurgent structures is variable, being characterized by elliptical domes with longitudinal graben(s) at the top (e.g. Toba, De Silva et al., 2015; Snowdonia, Beavon, 1980; Timber Mountain, Christiansen et al., 1977) or, less commonly, by sub-circular domes (e.g. Cerro Galán, Folkes et al., 2011; Long Valley, Hildreth et al., 2017; Grizzly Peak, Fridrich et al., 1991) with both longitudinal grabens (Long Valley) and concentric fault blocks (Grizzly Peak) at their top.

Whatever the shape, resurgence is often associated with hydrothermal and ore-forming processes, since the circulation pattern and temperature gradients of geothermal fluids are structurally controlled by the space–time distribution of faults and fractures and by the depth and shape of the magmatic sources (e.g. Guillou Frottier et al., 2000; Prinbow et al., 2003; Stix et al., 2003; Mueller et al., 2009; Giordano et al., 2014; Kennedy et al., 2018). Therefore, the characterization of the magma that drives resurgence (location, depth and size) and of the factors controlling the release of heat (permeability, fracture patterns and fluid flow) has important implications for the exploration and exploitation of renewable geothermal energy resources. In particular, the estimation of the location, depth and geometry of the magmatic sources is crucial to define the geothermal and mineral potential of resurgent calderas, allowing for an economically sustainable exploration and exploitation of their resulting natural resources.

The depth and size of magmatic sources influence the deformation style of the resurgence at the surface (Acocella et al., 2001). Deep sources (i.e. a depth-to-diameter ratio $\sim 1$ assuming a spherical source) are associated with resurgent blocks (e.g. Ischia and Pantelleria, Acocella and Funiciello, 1999; Catalano et al., 2009), whereas shallower sources (i.e. a depth-to-diameter ratio $\sim 0.4$) are associated with resurgent domes (e.g. Valles and Yenkahe, Kennedy et al., 2012; Brothelande et al., 2016). Moreover, uplift rates may change by 1 order of magnitude form $\sim 1$ to $\sim 10$ cm per year (e.g. Yellowstone and Iwo Jima, Chang et al., 2007; Ueda et al., 2018). Nevertheless, despite showing different uplift styles and rates, these natural examples share a common feature: a coherent uplift of the caldera floor. A different style of deformation is observed at calderas characterized by the widespread and delocalized uplift of several minor portions of the caldera floor, associated with the shallow emplacement of sills and cryptodomes, as observed at Usu volcano (Japan,

Matsumoto and Nakagawa, 2010; Tomya et al., 2010). Such a deformation pattern suggests different depth(s) and extent(s) of the magma source(s). A better assessment of the subsurface structure in this type of caldera has crucial implications for geothermal exploration.

The Los Humeros volcanic complex (LHVC; Mexico) is an important geothermal target area consisting of two nested calderas: Los Humeros (the outer, larger and older one; 164 ka) and Los Potreros (the inner, smaller and younger one; 69 ka) (Fig. 1). The latter is characterized by the resurgence of its floor, interpreted to be due to the inflation of the magma chamber responsible for the collapse, with its top at ca. 5 km of depth (Norini et al., 2015, 2019).

This paper aims to (1) evaluate the depth of the intrusion(s) inducing the uplift in the LHVC area; (2) explain the spatio-temporal evolution of the observed deformation of the caldera floor; and (3) test the validity of the linear relationship between the surface deformation structures and the depth of elliptical sources (Brothelande and Merle, 2015) for sub-circular sources. To achieve these goals, we integrate results from structural field investigations carried out within the Los Potreros caldera with those derived from analogue experiments specifically designed to constrain the depth of the deformation source(s) in volcanic caldera environments. The obtained results show the following: (1) the relation between the source depth and surface deformation structures is independent of the source eccentricity; and (2) the LHVC is characterized by discontinuous and small-scale (areal extent $\sim 1$ km$^2$) surface deformations generated from multiple shallow-emplaced ($< 1$ km depth) magmatic bodies. These results should be taken into account for the planning of future geothermal operations at the LHVC and in other calderas showing similar surface deformation.

## 2   Geological–structural setting

The LHVC is located at the eastern termination of the Trans-Mexican Volcanic Belt (TMVB; see inset in Fig. 1). The TMVB is the largest Neogene volcanic arc in Mexico ($\sim 1000$ km long and up to $\sim 300$ km wide), commonly associated with the subduction of the Cocos and Rivera plates beneath the North American Plate along the Middle American Trench (Ferrari et al., 2012, and references therein). The LHVC consists of two nested calderas formed during the Pleistocene: the outer $18 \times 16$ km Los Humeros caldera and the inner $10 \times 8$ km Los Potreros caldera (Fig. 1; Ferriz and Mahood, 1984; Norini et al., 2015; Carrasco-Núñez et al., 2017a).

Based on updated stratigraphic and geochronological information, the evolution of the LHVC can be divided into three main eruptive stages (Table 1; Carrasco-Núñez et al., 2017a, 2018). Pre-caldera volcanism extended between ca. 700 and 164 ka (zircon U–Th and feldspar $^{39}$Ar$/^{40}$Ar ages; Carrasco-Núñez et al., 2018), showing evidence for

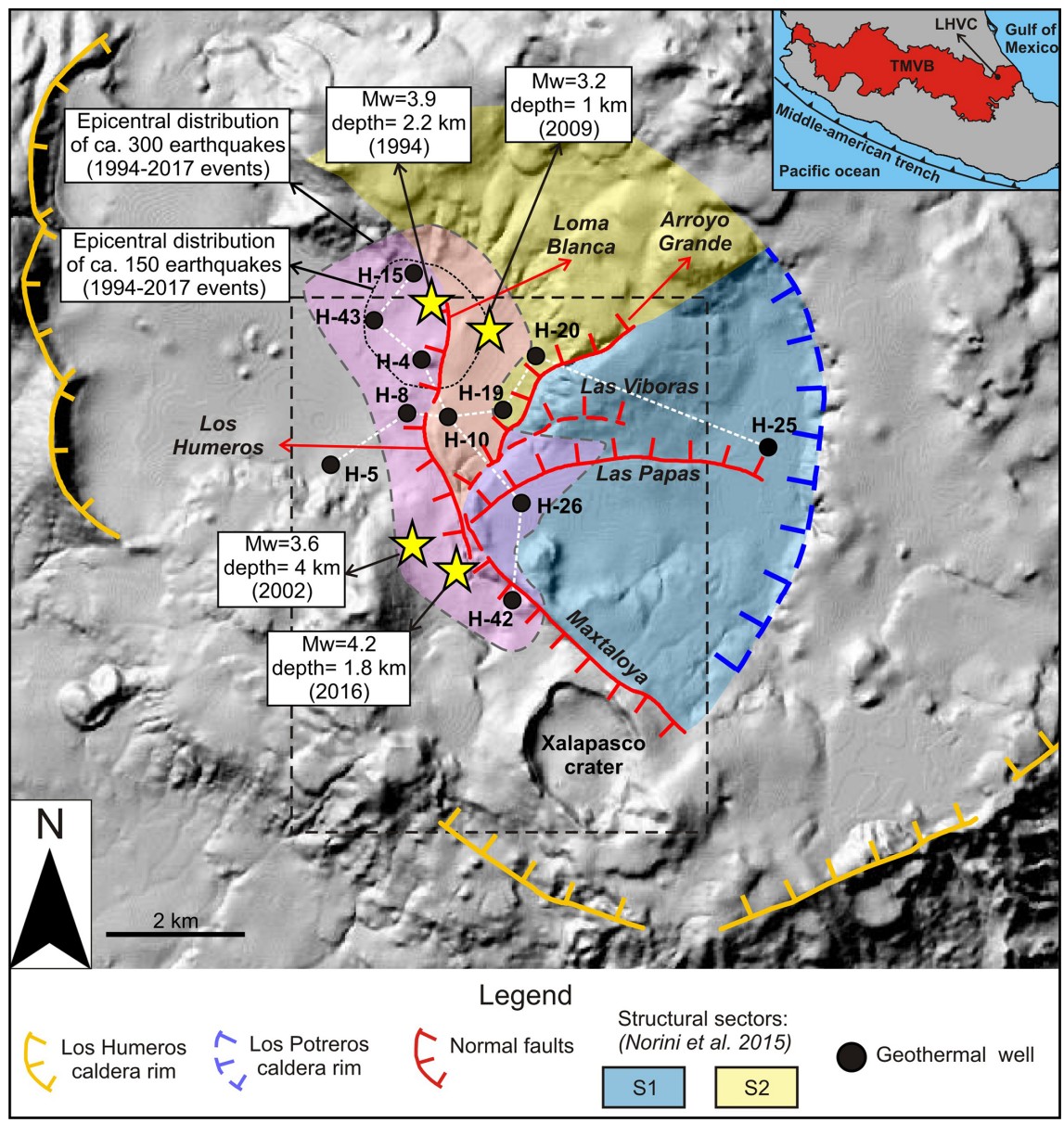

**Figure 1.** Shaded relief image (illuminated from the NE) obtained from a 15 m resolution digital elevation model (DEM) of the Los Humeros volcanic complex (LHVC) showing the main structural features (faults and caldera rim; modified from Norini et al., 2015; Calcagno et al., 2018) and some geothermal wells referred to in the text and in Figs. 2a–b. The white dashed lines indicate the direction of the correlation sections shown in Fig. 2a–b. The black rectangle indicates the studied area within the Los Potreros caldera shown in Fig. 4. The inset box shows the location of the LHVC (black dot and arrow) within the eastern sector of the Trans-Mexican Volcanic Belt (TMVB). The structural sectors S1 and S2 correspond to the resurgent block inferred by Norini et al. (2015). Seismicity data are from Lermo et al. (2018).

an extended building phase leading to the establishment of the large-volume rhyolitic reservoir, which fed several lava domes erupted to the western border of the Los Humeros caldera. A caldera stage started at ca. 164 ka (zircon U–Th and feldspar $^{39}$Ar/$^{40}$Ar ages; Carrasco-Núñez et al., 2018), with the eruption of the > 115 km$^3$ (dense rock equivalent volume) Xaltipan ignimbrite that triggered the collapse of the Los Humeros caldera. This was followed by a Plinian eruptive episodic sequence characterized by the emplacement of

several rhyodacitic pumice fallout layers grouped as the Faby Tuff (Ferriz and Mahood, 1984). The caldera stage ended with the eruption of the 15 km$^3$ (dense rock equivalent volume) Zaragoza rhyodacite–andesite ignimbrite at 69 ± 16 ka (feldspar $^{39}$Ar/$^{40}$Ar ages; Carrasco-Núñez et al., 2018) associated with the collapse of the nested Los Potreros caldera.

A post-caldera stage (< 69 ka) is interpreted by Carrasco-Núñez et al. (2018) as composed of two main eruptive phases: (i) a late Pleistocene resurgent phase characterized

**Table 1.** Summary of the main stratigraphic units of the three evolutionary stages of the Los Humeros volcanic complex (Carrasco-Núñez et al., 2017a, 2018).

| Stage | Age (ka) | Main stratigraphic units |
|---|---|---|
| Post-caldera | < 69 | Cuicuiltic Member and trachyandesitic to basaltic lavas<br>Llano Tuff<br>Xoxoctic Tuff<br>Rhyolitic domes |
| Caldera | 164–69 | Zaragoza ignimbrite<br>Faby Tuff<br>Xaltipan ignimbrite |
| Pre-caldera | 700–164 | Rhyolitic domes |

by the emplacement of silica-rich small domes and disperse explosive activity within Los Potreros caldera, followed by (ii) Holocene basaltic to trachytic monogenetic volcanism both inside and at the caldera rim. This eruptive behaviour indicates a change in the configuration of the magmatic plumbing system compared to the caldera stage of Los Humeros, when a single, large and homogenized magma reservoir was in existence (e.g. Ferriz and Mahood, 1984; Verma, 1985 TS4). Volcanological and petrological data indicate that the post-caldera volcanism is associated with a heterogeneous multilayered system vertically distributed within the crust, with a deep (ca. 30 km of depth) basaltic reservoir feeding progressively shallower and smaller distinct stagnation layers, pockets and batches up to very shallow conditions (ca. 3 km) (Lucci et al., 2020), in agreement with recent conceptual models for magma reservoirs under caldera systems (e.g. Cashman and Giordano, 2014).

During the early resurgent phase of the post-caldera stage, rhyolitic domes were emplaced along the northern rim and within the Los Humeros caldera. Available ages span between $44.8 \pm 1.7$ ka (zircon U–Th dating) and $50.7 \pm 4.4$ ka (feldspar $^{39}$Ar/$^{40}$Ar dating; Carrasco-Núñez et al., 2018). This effusive activity was followed by several explosive eruptions, which originated a dacitic air-fall called the Xoxoctic Tuff ($0.6$ km$^3$, Ferriz and Mahood, 1984) and a pyroclastic sequence that includes an explosive breccia and pyroclastic flow deposits comprising the Llano Tuff (Ferriz and Mahood, 1984; Willcox TS5, 2011).

The Holocene ring fractures fed bimodal magmatism characterized by both explosive and effusive activity, producing several lava flows and domes, as well as the ca. 7 ka (C-14 age; Dávila-Harris and Carrasco-Núñez, 2014) Cuicuiltic Member during periods of dominant explosive activity. The Cuicuiltic Member consists of alternating pumices and scoriae erupted during contemporaneous sub-Plinian to Strombolian activity from multiple vents located mostly along the inner part of the caldera and outer caldera ring faults (Dávila-Harris and Carrasco-Núñez, 2014). During this phase, less evolved lavas (trachyandesite to basalt) were erupted within and outside the Los Potreros caldera, including the olivine-

bearing basaltic lava that fills the previously formed Xalapasco crater (Fig. 1). Trachytic lava flows are the most recent products in the area, with an age of ca. 2.8 ka (C-14 age; Carrasco-Núñez et al., 2017a).

The reconstruction of the shallow stratigraphy within the Los Potreros caldera is chiefly derived from the analysis of available well logs (Fig. 2a–b; Carrasco-Núñez et al., 2017a, b). Overall, the post-caldera units are lithologically dominated by lava flows resting on ignimbrite deposits emplaced during the caldera stage. Ignimbrites of the caldera stage rest in turn on a thick sequence dominated by andesite lavas dated at ca. 1.4–2.8 Ma (feldspar $^{39}$Ar/$^{40}$Ar dating; Carrasco-Núñez et al., 2017b). The subsurface geometry of the pre- and syn-caldera products is shown in Fig. 2a–b, where the in-depth geometry of the different magmatic products is cross-correlated and projected along the N–S and E–W direction, respectively. The N–S projection shows a constant depth of the top surface of the pre-caldera andesites that is associated with a highly variable depth (down to $-400$ m) of the top surface of the syn-caldera Xaltipan ignimbrite. The W–E projection shows a higher depth variability of both the top surface of the pre-caldera group (down to $-500$ m between H-19 and H-25 wells) and that of the Xaltipan ignimbrite (down to $-400$ m between H-19 and H-10 wells). Basaltic and rhyolitic–dacitic lavas occur at various depths (Carrasco-Núñez et al., 2017b); rhyolites–dacites are located mostly at the base (H-20 and H-26 wells) or within (H-05 well) the caldera group or the old andesite sequence (H-25 and H-19 wells). Basalts are located only within the pre-caldera andesite sequence, both at its base (in contact with the limestone basement; H-5 and H-8 wells) and at its top (in contact with the base of the caldera sequence; H-10 well). These bimodal lava products, showing an irregular lateral distribution, have been interpreted as subaerial (Carrasco-Núñez et al., 2017b).

The structural architecture of the LHVC is controlled by a network of active extensional fault systems made of NNW–SSE-, N–S-, NE–SW- and E–W-striking fault strands cutting across the Los Potreros caldera floor. The following main faults were recognized (Norini et al., 2015, 2019; Calcagno

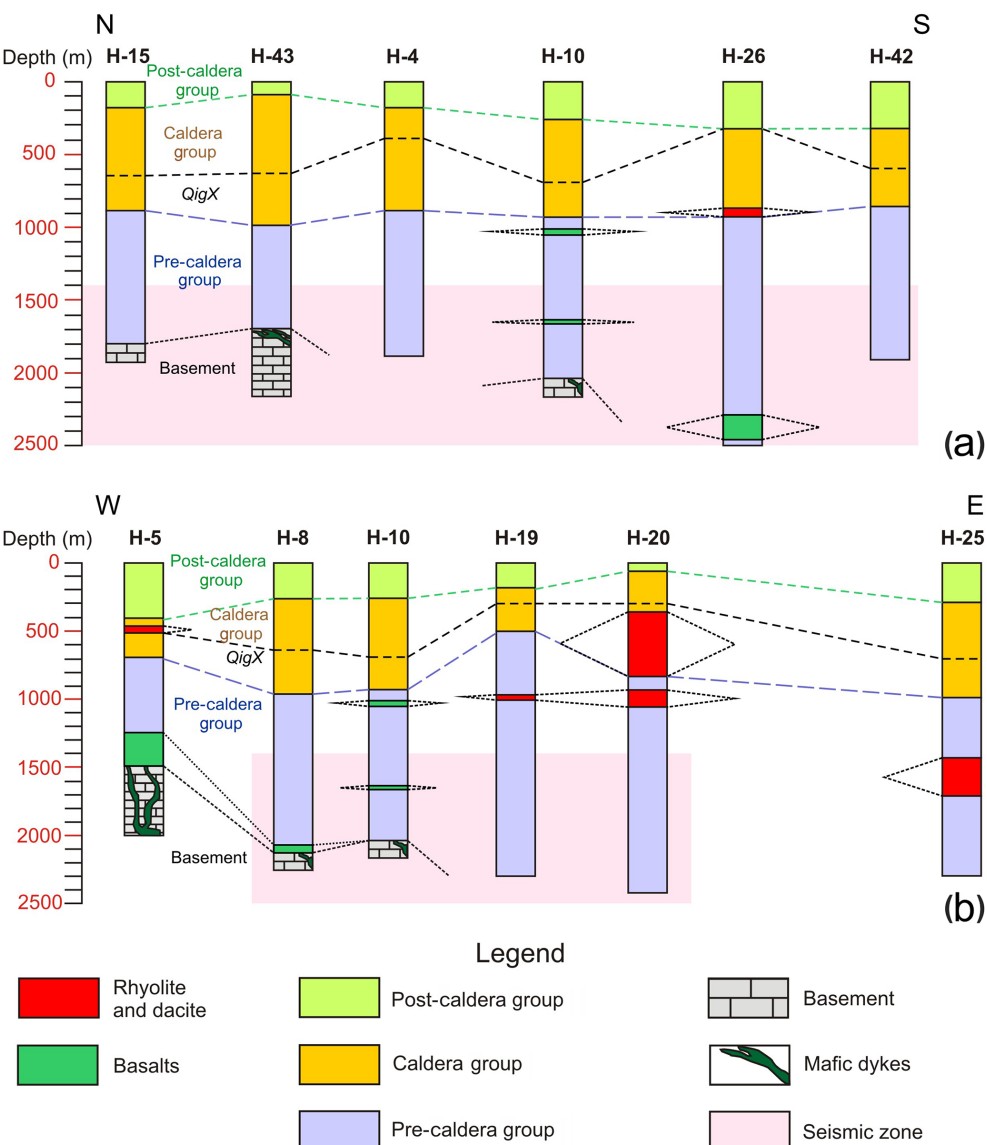

**Figure 2.** In-depth correlation of lithostratigraphic units along the N–S **(a)** and W–E **(b)** direction; redrawn after Carrasco-Núñez et al., (2017a) and Arellano et al. (2003). The ratio of depth to horizontal distance is 1 : 1. The location of the correlation line is shown in Fig. 1. QigX: Xaltipan ignimbrite.

et al., 2018) (Fig. 1): (i) Maxtaloya (NNW–SSE striking), (ii) Los Humeros and Loma Blanca (N–S striking), (iii) Arroyo Grande (NE–SW striking), and (iv) Las Viboras and Las Papas (E–W striking). Such active fault systems are interpreted as due to the recent active resurgence of the Los Potreros caldera. Since the faults do not show continuity beyond the caldera border, their scarps decrease in height towards the periphery of the caldera, and the dip-slip displacement vectors show a semi-radial pattern (Norini et al., 2015).

The source of the areal uplift has been inferred to be the inflation of a saucer- or cup-shaped deep magmatic source elongated NNW–SSE, upwarping a $8 \times 4$ km resurgent block, centred in the SE portion of the caldera, delimited to the W by the NNW–SSE main faults and toward the north, east and south by the caldera rim (Fig. 1; Norini et al., 2015, 2019).

The seismic activity between 1994 and 2017 is clustered along the Loma Blanca, Los Humeros and Arroyo Grande faults (Lermo et al., 2018; Fig. 1). Most of the earthquakes show a magnitude ($M_w$) between 1 and 2.5 and have been mainly interpreted as induced by geothermal exploitation activity (injection of fluids and hydrofracturing; Lermo et al., 2018). Four major earthquakes ($M_w = 3.2$, 3.6, 3.9 and 4.2 at depths of 1, 4, 2.2 and 1.8 km, respectively) have also been reported, with focal depths close to the trace of the active faults (Loma Blanca and Los Humeros; Fig. 1). Such major earthquakes have been interpreted as triggered by fault reac-

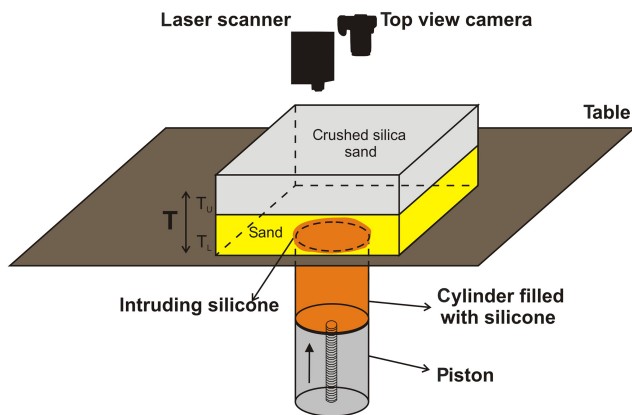

**Figure 3.** Experimental set-up. A motor-controlled piston pushes the silicone upward at a fixed rate ($2\,\mathrm{mm\,h^{-1}}$) from the base of the layered sand pack (the diameter of the silicone is 8 cm). A laser scanner and a camera record the surface deformation induced by the intruding silicone. $T$: total overburden thickness. $T_\mathrm{u}$: upper layer thickness, $T_\mathrm{l}$: lower layer thickness.

tivation due to fluid–brine circulation injected from geothermal wells (Lermo et al., 2018).

## 3   Methods

This study is based on structural fieldwork combined with analogue models aimed at constraining the depth of the deformation sources in the caldera domain. We also tested if the relation that constrains the depth of the source deformation from surface parameters adopting elliptical sources (Brothelande and Merle, 2015) is also verified for sub-circular sources.

### 3.1   Structural fieldwork

Structural fieldwork was carried out on the post-caldera (late Pleistocene to Holocene) deposits to characterize the surface deformation related to the recent activity of the Los Potreros caldera and constrain the morphotectonic fingerprints of the resurgence to evaluate its source and areal extent. The geometry and distribution of the observable faults and joints were defined at the outcrop scale by measuring their attitudes (strike and dip; right-hand rule) and spacing. Fault kinematics were assessed through classical criteria on slickensides fault surfaces, such as Riedel shears, growth fibres and sheltering trails (Doblas, 1998). The published geological map (Carrasco-Núñez et al., 2017a) and geothermal well data have been used (Carrasco-Núñez et al., 2017b) to correlate the surface structures at a broader scale. The relationships between faulting and alteration have been assessed (e.g. Giordano et al., 2013; Vignaroli et al., 2013, 2015).

## 3.2   Analogue models: experimental set-up and scaling

Five experiments were undertaken to simulate the ascent of a viscous sub-circular intrusion in a brittle overburden to test the validity of existing relationships between the depth of elliptical intrusions and the observed surface deformation (Brothelande and Merle, 2015). The experimental set-up (Fig. 3) consists of a $31 \times 31$ cm glass box filled with a sand pack (crust analogue) of variable thickness ($T$, of 10, 30 and 50 mm). In each experiment we imposed a layering using a non-cohesive marine sand below a layer of crushed silica sand (grain size 40–200 µm, cohesion 300 Pa), fixing the thickness ratio of the two layers ($T_\mathrm{u}/T_\mathrm{l}$) to 1, to simulate the stratigraphy in Los Potreros (stiffer post-caldera lava flows above softer and less cohesive ignimbrite deposits emplaced during the caldera collapse stage). At the base of the sand pack, a piston, controlled by a motor, pushes the silicone upward (magma analogue) inside a cylinder 8 cm in diameter. The injection rate is fixed for all the experiments to $2\,\mathrm{mm\,h^{-1}}$, TS6 and each experiment was stopped at the onset of the silicone extrusion. Both the sand and silicone physical properties are listed in Table 2.

At the end of each experiment, the surface was covered with sand to preserve the final topography and wetted with water for cutting in sections to appreciate the subsurface deformation. Such sections were used to measure the mean dip of the apical depression faults ($\theta$) induced by the rising silicone. A digital camera monitored the top view deformation of each experiment at 0.02 fps, and a laser scanner, placed next to the camera, provided high-resolution data (maximum error $\pm 0.5$ mm) of the vertical displacement to measure in detail the geometrical features of the deformation, i.e. dome diameter ($L_\mathrm{d}$), apical depression width ($L_\mathrm{g}$) and dome flank mean dip ($\alpha$). According to the Buckingham $\Pi$ theorem (Merle and Borgia, 1996, and references therein), our models need seven independent dimensionless numbers to be properly scaled (i.e. 10 variables minus three dimensions; Table 2). Such dimensionless numbers can be defined as the ratios ($\Pi$) listed in Table 3. Some values of $\Pi_5$, representing the ratio between the inertial and viscous forces, are very small both in nature and the experiments ($1.3 \times 10^{-20}$ and $6.1 \times 10^{-10}$, respectively), indicating that the inertial forces are negligible compared to the viscous forces in both cases.

## 4   Results

### 4.1   Structural geology

The outcropping post-caldera lithologies within the Los Potreros caldera consist of the following: (1) the Cuicuiltic Member, which blankets most of the surface of the upper half of the studied area; (2) basaltic lava flows filling the Xalapasco crater and the NW portion of the caldera; and (3) trachyandesitic and trachytic lava domes and thick flows

**Table 2.** Comparison of the geometric and material property parameters of the experiments and nature.

| Parameter | Definition | Value (experiments) | Value (nature) |
|---|---|---|---|
| $T$ | Thickness of the overburden | $1$–$5 \times 10^{-2}$ m | $300$–$2000$ m |
| $L_d$ | Dome diameter | $1$–$1.6 \times 10^{-1}$ m | $2000$ m |
| $H$ | Dome height | $1.1$–$2 \times 10^{-2}$ m | $100$ m |
| $\rho_s$ | Density of brittle overburden | $1400\,\mathrm{kg\,m^{-3}}$ | $2800\,\mathrm{kg\,m^{-3}}$ |
| $\phi$ | Angle of internal friction | $35°$ | $25$–$40°$ |
| $\tau_0$ | Cohesion (brittle overburden) | $300$ Pa | $10^6$ Pa |
| $\rho_m$ | Density of intrusive material | $1000\,\mathrm{kg\,m^{-3}}$ | $2500\,\mathrm{kg\,m^{-3}}$ |
| $\mu_m$ | Viscosity of intrusive material | $10^4$ Pa s | $10^{15}$ Pa s |
| $g$ | Gravity | $9.8\,\mathrm{m\,s^{-2}}$ | $9.8\,\mathrm{m\,s^{-2}}$ |
| $t$ | Time span for deformation | $2.8$–$6.5 \times 10^4$ s | $1.9 \times 10^{12}$ s |

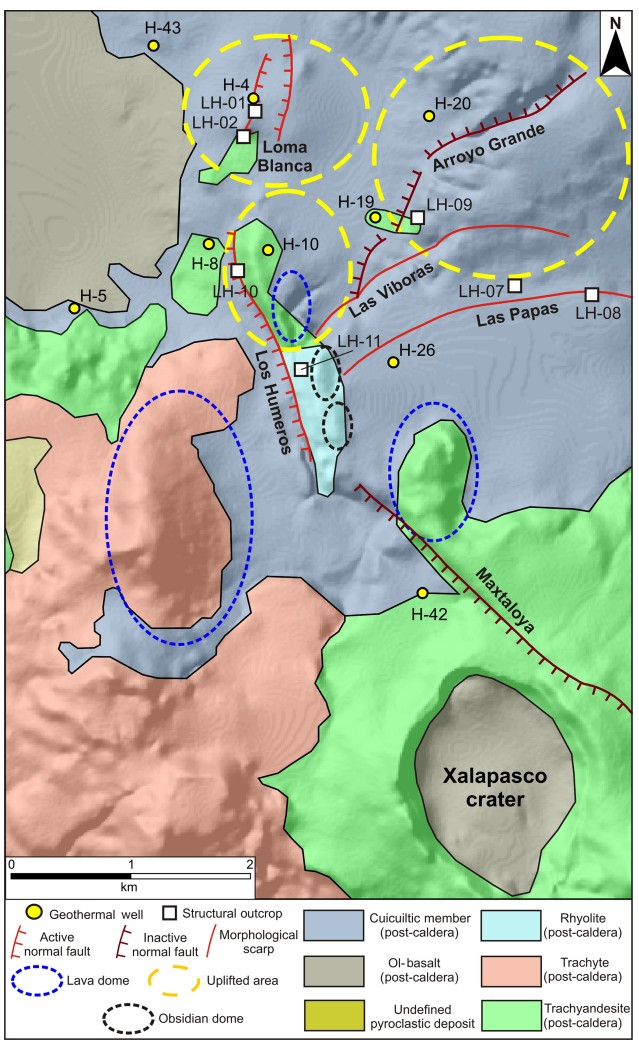

**Figure 4.** Simplified geological structural map of the studied area (reinterpreted after Norini et al., 2015; Carrasco-Núñez et al., 2017a; Calcagno et al., 2018).

**Table 3.** Definition and values of the dimensionless ratios $\Pi$ in nature and in the experiments.

| Dimensionless ratio | Experiments | Nature |
|---|---|---|
| $\Pi_1 = T/L_d$ | $0.1$–$0.5$ | $0.15$–$1$ |
| $\Pi_2 = H/L_d$ | $0.08$–$0.2$ | $0.05$–$0.1$ |
| $\Pi_3 = \rho_s/\rho_m$ | $1.4$ | $1.12$ |
| $\Pi_4 = \phi$ | $35$ | $25$–$40$ |
| $\Pi_5 = \rho_m H^2/\mu_m t$ | $6.1 \times 10^{-10}$ | $1.3 \times 10^{-20}$ |
| $\Pi_6 = \rho_m g H t/\mu_m$ | $1.3 \times 10^3$ | $4.6 \times 10^3$ |
| $\Pi_7 = \rho_s g T/\tau_0$ | $2.3$ | $8.24$ |

extending in the southern half of the caldera, with rhyolitic domes in its central part (Fig. 4). Fieldwork documented that the more evolved lavas form five nearly N–S-trending elliptical domes, distributed on both sides of the Los Humeros Fault (Figs. 4 and 5a): (i) a 2 km long × 1.2 km wide trachytic dome located to the west of the Maxtaloya and Los Humeros faults, (ii) a 1 × 0.7 km trachyandesitic dome located in a north-east area of the Maxtaloya fault, and (iii) one trachyandesitic and two obsidian smaller domes (0.4 × 0.2 km) to the eastern side of the Los Humeros Fault (LH-11 in Fig. 4).

Fieldwork concentrated on the three main uplifted areas corresponding to the surface expression of the Loma Blanca, Arroyo Grande and Los Humeros faults (labelled LH1-2, LH9 and LH10, respectively, in Fig. 4). The observed structures in these uplifted areas (joints and faults) affect the deposits of the post-caldera phase. Based on field evidence, we also propose a revised interpretation of the surface structures identified by previous studies (Norini et al., 2015, 2019), distinguishing between lineaments (morphological linear scarps with no measurable fault offsets and/or alteration at the outcrop scale) and active and inactive faults that are instead associated with measurable fault offsets and with active or fossil alteration, respectively (Fig. 4). We detail below the main structures mapped in the studied area, highlighting their temporal and spatial relationships with the post-caldera geological formations. We identified two inactive faults (Maxtaloya

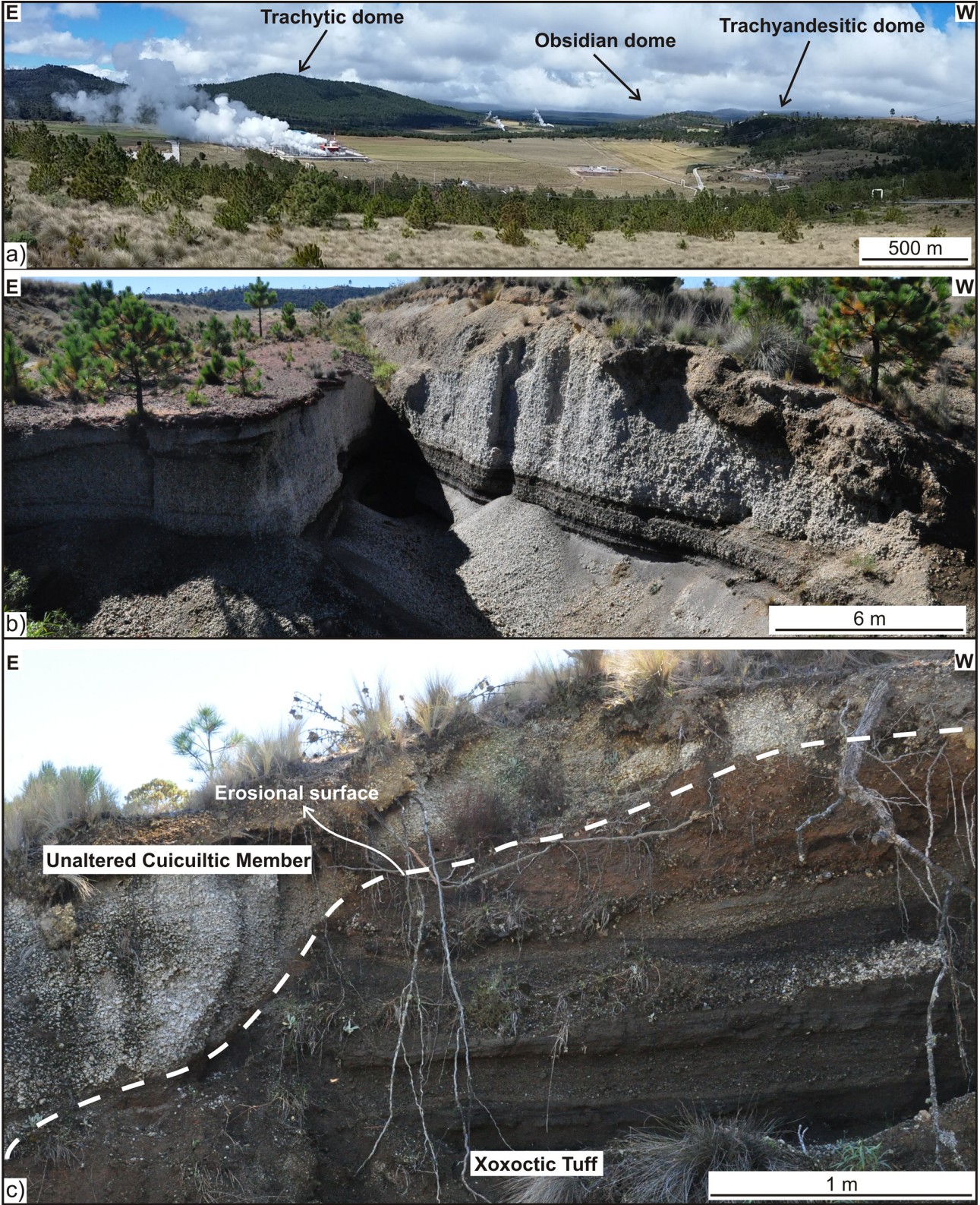

**Figure 5. (a)** Panoramic view from Xalapasco crater (looking towards N) of the lava domes aligned N–S. **(b)** Unaltered Cuicuiltic Member (LH-07). **(c)** Unaltered Cuicuiltic Member covering a layered pyroclastic deposit, which can be laterally correlated with the Xoxoctic Tuff (LH-08). The erosional surface preceding the deposition of the Cuicuiltic Member is shown (dashed white line).

and Arroyo Grande), a morphological lineament (Las Papas) and two currently active faults (Los Humeros and Loma Blanca).

### 4.1.1 Las Papas lineament (sites LH-07, LH-08)

The E–W-trending Las Papas lineament is localized within the Cuicuiltic Member (LH-07; Fig. 5b). We identified an erosional surface along the scarp, where unaltered and undeformed Cuicuiltic Member rocks rest above the Xoxoctic Tuff (LH-08, Fig. 5c). The E–W-trending morphological lineament of Las Papas is probably due to differential erosion of the softer layers of the pyroclastic deposits, successively blanketed by the Cuicuiltic Member.

### 4.1.2 Arroyo Grande (site LH-09) and Maxtaloya scarps

The NE–SW Arroyo Grande scarp (Fig. 6a) exposes strongly altered and faulted (NW-striking faults, mean attitude 144/68° N, number of data points $n = 8$) lavas and ignimbrites unconformably covered by the unaltered Cuicuiltic Member (Fig. 6b). The offset observed at the outcrop scale for the single fault strands is ca. 0.5 m, with dominant normal dip-slip kinematics (the pitch angle of the slickenlines ranges 99–106°). The inferred cumulative displacement at Arroyo Grande is $\sim 10$ m. Similarly, an outcrop on the Maxtaloya scarp (in front of well H-6) shows altered trachyandesites covered by unaltered Cuicuiltic Member rocks (Fig. 6c).

### 4.1.3 Los Humeros (site LH-10)

The fault scarp of the N–S-striking (mean attitude 174/73° N, $n = 8$) Los Humeros Fault exposes the altered portions of the Cuicuiltic Member. Fault population analysis reveals dominant normal dip-slip (mean pitch angle of the slickenlines: 84°) kinematics, as documented by both Riedel shears and carbonate–quartz growth steps. The main fault surface is sutured by a trachyandesitic extrusion (Fig. 6d) localized along an aligned N–S dome (site LH-11 in Fig. 4). Moreover, $\sim 150$ m southward from the outcrop of the fault scarp, a $5 \times 3$ m wide trachyandesitic plug shows vertical striation on its surface due to a subsurface vertical flow of the trachyandesite (Fig. 6e). The observed displacement at the outcrop scale, as indicated by the height of the fault scarp, is $\sim 10$ m.

### 4.1.4 Loma Blanca (LH-01, LH-02)

The Loma Blanca Fault system (sites LH-01 and LH-02) is located in an active degassing area, where faults and fractures are frequent. The fault system is on top of an elongated crest (within an apical depression) of a morphological bulge, $\sim 1$ km in width and 30 m in height. At this location, the Cuicuiltic Member and the underlying trachyandesite lavas are strongly altered (Fig. 6f). Evidence of stockwork veining and diffuse fracturing of the lavas suggests hydrofrac-

**Table 4.** Measured ($L_g$, $L_d$, $\theta$, $\alpha$) and imposed ($T$) parameters in the experiments. $T$: overburden thickness; $L_d$: dome diameter; $L_g$: apical depression width; $\theta$: apical depression fault dip; $\alpha$: dome flank mean dip; $T_t$: theoretical overburden thickness calculated with Eq. (1) (Brothelande and Merle, 2015; see the Discussion section); $\sigma$: percentage difference between $T$ and $T_t$.

| Exp. | $T$ (mm) | $L_g$ (mm) | $L_d$ (mm) | $\theta$ | $\alpha$ | $T_t$ (mm) | $\sigma$ (%) |
|---|---|---|---|---|---|---|---|
| 1 | 10 | 16 | 116 | 58° | 14° | 15.5 | 55 |
| 2 | 10 | 14 | 105 | 63° | 27° | 15.4 | 54 |
| 3 | 30 | 42 | 150 | 58° | 14° | 37.7 | 27 |
| 4 | 30 | 48 | 138 | 56° | 18° | 41.2 | 37 |
| 5 | 50 | 58 | 164 | 58° | 21° | 53.7 | 7 |

turing and structurally controlled fluid flow and alteration. A set of NNE–SSW-striking conjugate extensional faulting and jointing (joint spacing $\sim 0.5$ m) is observed. The faults (mean attitude 26/71° N, $n = 6$) show normal dip-slip kinematics (the pitch of the slickenlines ranges 82–104°). Joint systems found in the Cuicuiltic Member strike subparallel to the faults (mean attitude 37/72° N, $n = 14$). The inferred cumulative displacement of the faults, estimated by the depth of the apical depression, is $\sim 5$ m.

In summary, the 22 mapped faults in all the structural outcrops of the area show a main NNW–SSE strike (Fig. 6g) with a dominant dip-slip movement (mean pitch angle of slickenlines 88°, $n = 16$) which is subparallel to the N–S elongation of the lava domes and the Xalapasco crater.

## 4.2 Experimental results

Here we show three representative experiments with increasing overburden thickness (experiments 1–3–5 with $T = 10$, 30 and 50 mm). Table 4 shows the measured parameters in the experiments. Some experiments (1–2 and 3–4) were replicated with the same imposed boundary conditions and show the same result (i.e. apical depression width and dome diameter), which ensures model reproducibility (Figs. 8 and S1 in the Supplement).

Overall, the experiments show a similar deformation pattern: a first stage characterized by the uplift of a sub-circular dome bordered by inward-dipping reverse faults and a second stage characterized by the subsidence of the apical part of the dome where normal faulting occurs (apical depression formation; Fig. 7a–i). The reverse and normal faults are ring faults and are associated with the formation of radial fractures from the dome centre. A different shape of the apical depression is observed with $T/D > 0.12$. In exp. 1 ($T/D = 0.12$) an annular peripheral depression formed as the silicone reached the surface at the edge of the cylinder (Fig. 7c). Conversely, in exps. 3 and 5 ($T/D = 0.37$ and 0.63, respectively) a sub-circular apical depression formed

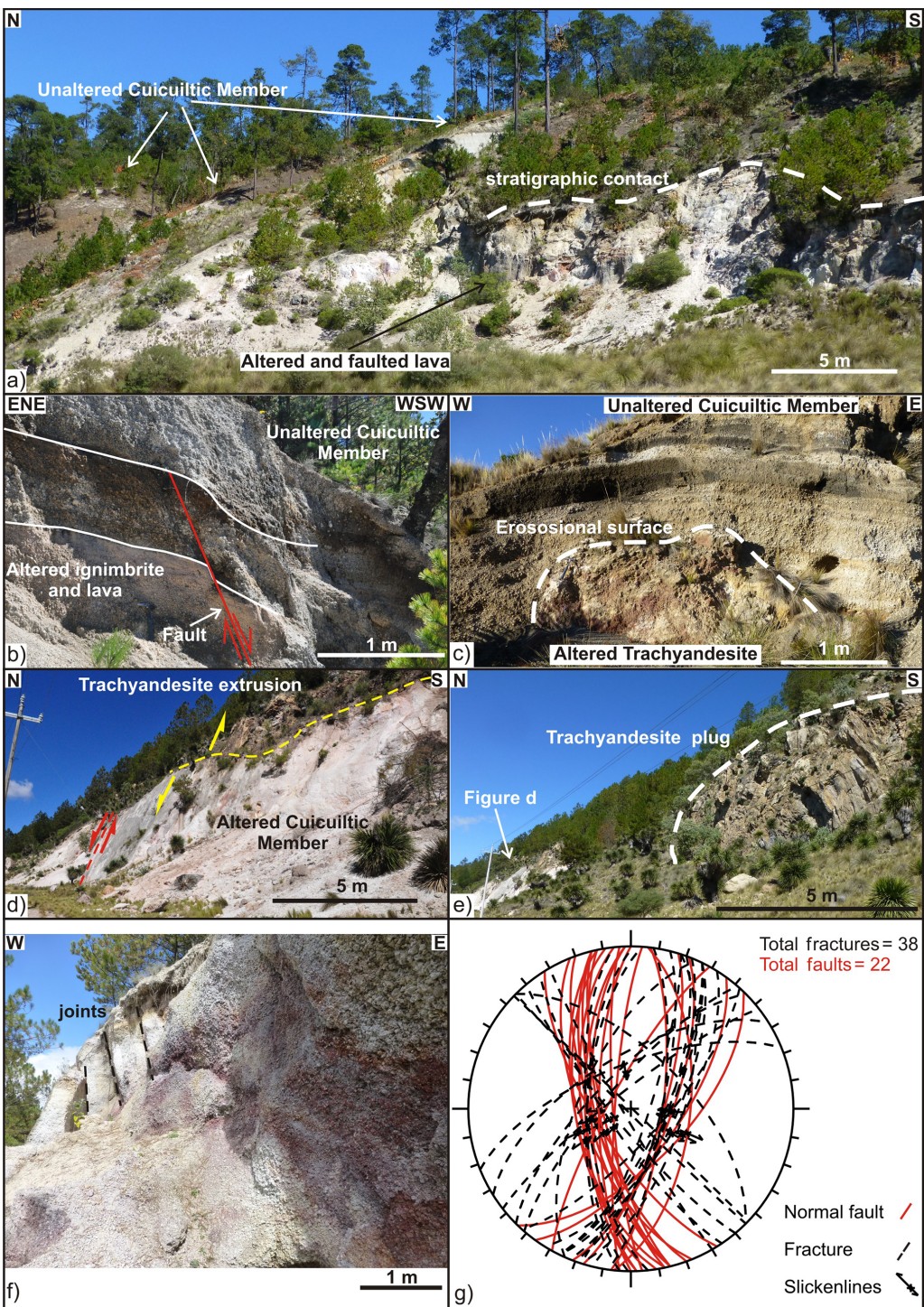

**Figure 6. (a)** Panoramic view of the Arroyo Grande fault scarp showing the unaltered Cuicuiltic Member covering the altered and faulted ignimbrite and lavas (site LH-09). **(b)** Normal fault affecting the altered ignimbrite deposits unconformably covered by the post-caldera, unaltered Cuicuiltic Member deposits (LH-09). Note that the Cuicuiltic Member deposits are not faulted at this location; the fault can thus be considered a fossil fault with respect to the Cuicuiltic Member deposition. **(c)** Block of altered trachyandesite buried by unaltered Cuicuiltic Member layers along the Maxtaloya fault scarp. **(d)** Los Humeros Fault scarp (LH-10) induced by the ascent of the trachyandesitic extrusion on top of the fault plane. **(e)** Trachyandesite plug cropping out ∼ 150 southward of the fault scarp shown in **(d)** (indicated by the red arrow). **(f)** Jointing and alteration of the Cuicuiltic Member within the apical depression of the Loma Blanca dome (LH-01). **(e)**TS7 Equal-area stereo plot of the attitudes of faults and fractures in all the structural outcrops.

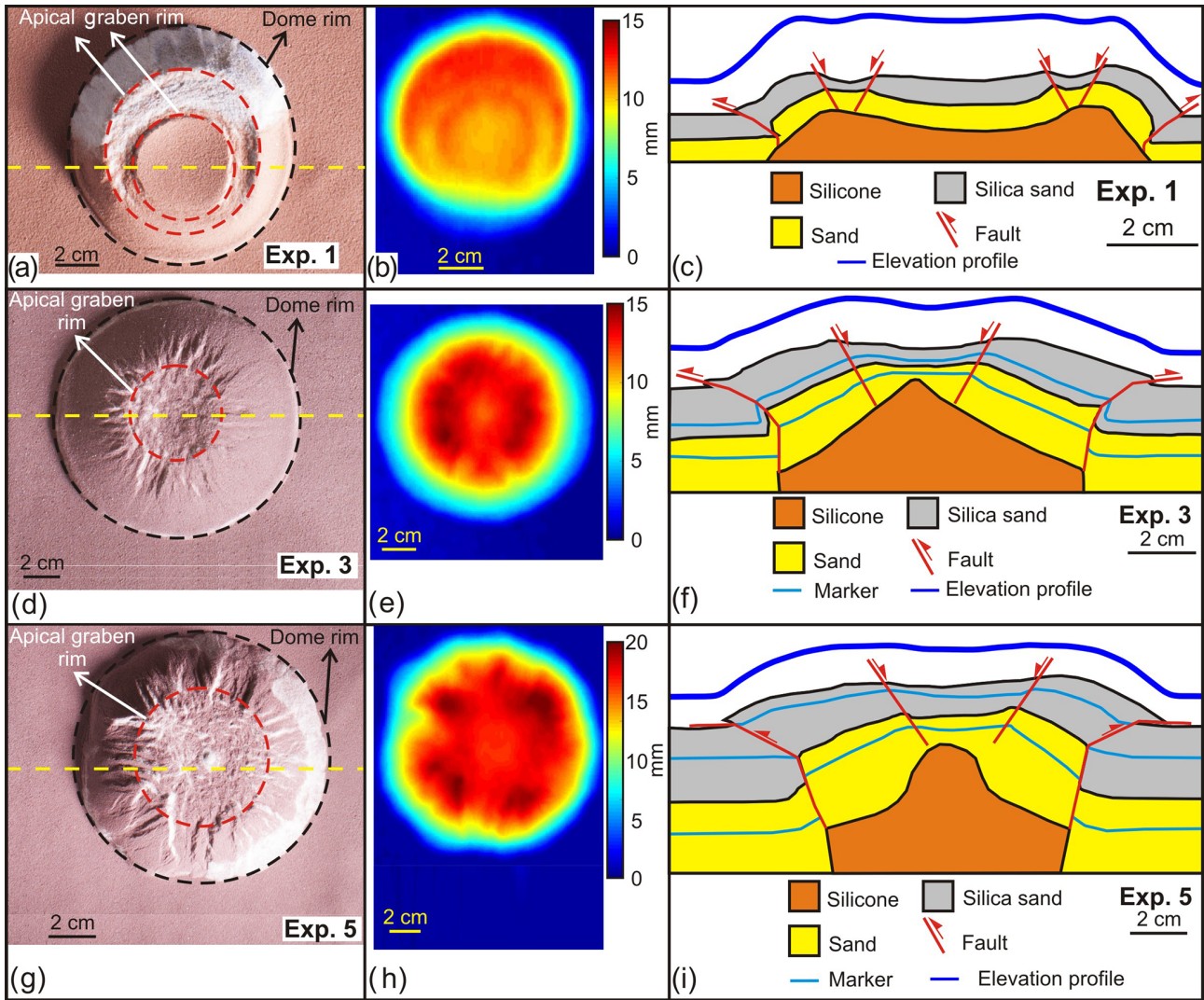

**Figure 7. (a)**, **(d)**, **(g)** Top view image of experiments 1, 3 and 5. **(b)**, **(e)**, **(h)** Cumulative vertical displacement; the colour scale is proportional to the amount of uplift. **(c)**, **(f)**, **(i)** Drawing of the cross section view obtained after cutting the section close to the dome centre. The elevation profiles are obtained from laser scanner data. The yellow dashed line in **(a)**, **(d)** and **(g)** indicates the trace of the section views and of the elevation profiles.

as the silicone reached the surface at the centre of the dome (Fig. 7g, m).

Irrespective of the $T/D$ ratio, all experiments show that both the dome diameter and apical depression width increase linearly with the overburden thickness (ranging from 105 to 164 and 14 to 58 mm, respectively; Table 4, Fig. 8). The dome diameter increases abruptly with time, becoming almost constant at an early stage of the experiment (Fig. 9a); the apical depression width shows a similar pattern even if it enlarges slightly with time (after the first abrupt increase) as the silicone rises towards the surface (Fig. 9b), suggesting that the intrusion depth has a higher influence on the apical depression width, in agreement with Brothelande and Merle (2015).

## 5 Discussion

### 5.1 Interpretation of the analogue experiments

The deformation pattern observed in the analogue experiments for thicker overburdens (experiments 3–4 and 5 with $T/D = 0.37$ and 0.63), showing a sub-circular dome and an apical depression, is in agreement with previous analogue experimental results (Acocella et al., 2001; Martì et al., 1994; Walter and Troll, 2001). However, for thinner overburdens (exps. 1–2, $T/D = 0.12$), we observed a new deformation pattern at the surface consisting of an annular peripheral depression due to the rising of the silicone at the edge of the cylinder rather than its centre. We infer that in these experiments, since the rising silicone was very close to the sur-

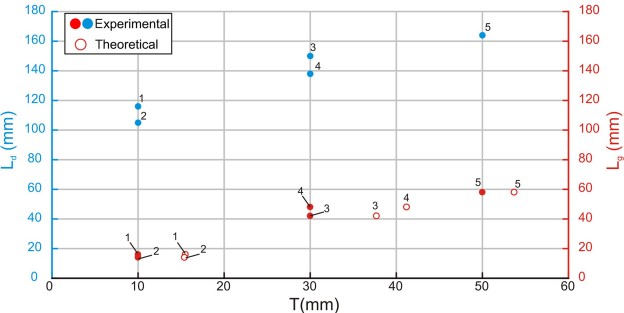

**Figure 8.** $L_g$ (apical depression width) and $L_d$ (dome diameter) versus $T$ (overburden thickness). Theoretical values calculated after Eq. (1) (see the Discussion section). The numbers above each point indicate the experiment number.

face, the sagging of the sand overburden pushed the silicone downward, which consequently squeezed up at the edges of the cylinder. Such a process may also explain the two linear grabens that formed in the experiments with elliptical sources for small overburden thicknesses (ratio $T/D \sim 0.1$; Brothelande and Merle, 2015).

The deformation pattern observed in our experiments is independent of the imposed strain (i.e. uplift) rate or the viscosity of the intruding material as suggested by the similarity with results obtained in previous studies with higher strain rates (Acocella and Mulugeta, 2002) or lower-viscosity intruding materials (Galetto et al., 2017; Martì et al., 1994; Walter and Troll, 2001). On the other hand, the occurrence of an apical depression is dependent on the thickness (i.e. depth) of the intrusion since thin intrusions relative to their depths will generate sub-circular domes without any apical depression (Galland et al., 2009; Galland, 2012). Moreover, our results confirm that the apical depression width shows a linear correlation with the source depth (Fig. 8) as estimated in Brothelande and Merle (2015) for elongated sources. This evidence documents that such a relation is independent of the source eccentricity or shape of the extensional structures at the top of the dome (i.e. linear graben or sub-circular depression), suggesting that any elongation of the surface structure represents only a minor complication of the basic deformation pattern as already pointed out by Roche et al. (2000).

## 5.2 Origin and extent of the resurgence in the LHVC

The distribution of alteration patterns and deformation characteristics of the post-caldera deposits can be used to infer the origin and extent of the uplift within the Los Potreros resurgent caldera. In particular, we focus on the Holocene Cuicuiltic Member, which blankets the caldera floor. Unaltered and undeformed deposits of the Cuicuiltic Member crop out along the E–W Las Papas lineament and unconformably cover altered and faulted lavas and ignimbrites along the Arroyo Grande and Maxtaloya scarps. Alteration and deformation of the Cuicuiltic Member occur along the

Los Humeros Fault scarp and within the apical depression of the Loma Blanca bulge. The vertical striations of the trachyandesitic plug near the Los Humeros Fault scarp suggest that the ascent of the plug induced the uplift, the normal dipslip faulting and alteration of the Cuicuiltic Member.

The observations suggest that Los Potreros is not a classic resurgent caldera (i.e. a caldera characterized by a large-scale process localized in a single area) but is characterized by uplift pulses discontinuous in space and time, inducing small-scale deformations at each pulse (Fig. 10a–d). In particular, it was active in the south and north-eastern sector of the caldera, at Maxtaloya and Arroyo Grande (Fig. 10a), prior to the deposition of the Cuicuiltic Member ($\sim$ 7.4 ka), and then shifted north along the Los Humeros and Loma Blanca scarps during and after the eruption of the Cuicuiltic Member (Fig. 10b–d). The felsic lava found at the Los Humeros Fault scarp shows a similar mineral assemblage as the felsic domes located further south (Fig. 4); thus, the Los Humeros scarp may represent the final stage (i.e. effusive eruption of felsic magmas; Fig. 10c) of the uplift process, which is thus driven by the ascent of relatively narrow (hundreds of metres) and highly viscous felsic magma batches. This is supported by the N–S elongation of the identified lava domes, which is subparallel to the orientation of the measured fault planes (NNW–SSE), indicating that the observed deformation is closely related to the post-caldera volcanism. The emplacement of such magma bodies is inferred here to drive the recent uplift and deformation of the Loma Blanca bulge, as suggested by the active fumaroles and extensive alteration of both the Cuicuiltic Member and post-caldera lavas (Fig. 10d). The recent emplacement of shallow magma bodies should be considered a possible scenario for the interpretation of the seismicity in Los Potreros, which has so far been interpreted as induced by geothermal exploitation (Lermo et al., 2018). In fact, the highest magnitude of the recent seismicity reached between 3.2 and 4.2 and may well be consistent with a volcano-tectonic origin due to shallow magma emplacement, rather than induced by the reinjection of hydrothermal fluids (see Evans et al., 2012, and references therein).

To further support the above interpretation from field observations, results from the presented analogue models were used to constrain the magma source depth from the geometrical parameters measured in the experiments ($L_g$, $\theta$, $\alpha$; Table 4). We calculated the theoretical overburden thickness (i.e. the intrusion depth, $T_t$; Table 4) as follows (Brothelande and Merle, 2015).

$$T_t = \frac{1}{2} L_g \times \frac{\sin(\theta + \alpha)}{\cos\theta} \quad (1)$$

Comparing the percentage difference ($\sigma$) between the imposed experimental ($T$) and theoretical ($T_t$) overburden thickness values, we calculate the associated error in the evaluation of the intrusion depth in the models ($\sigma$; Table 4, Fig. 8). We then use Eq. (1) for the evaluation of the heat

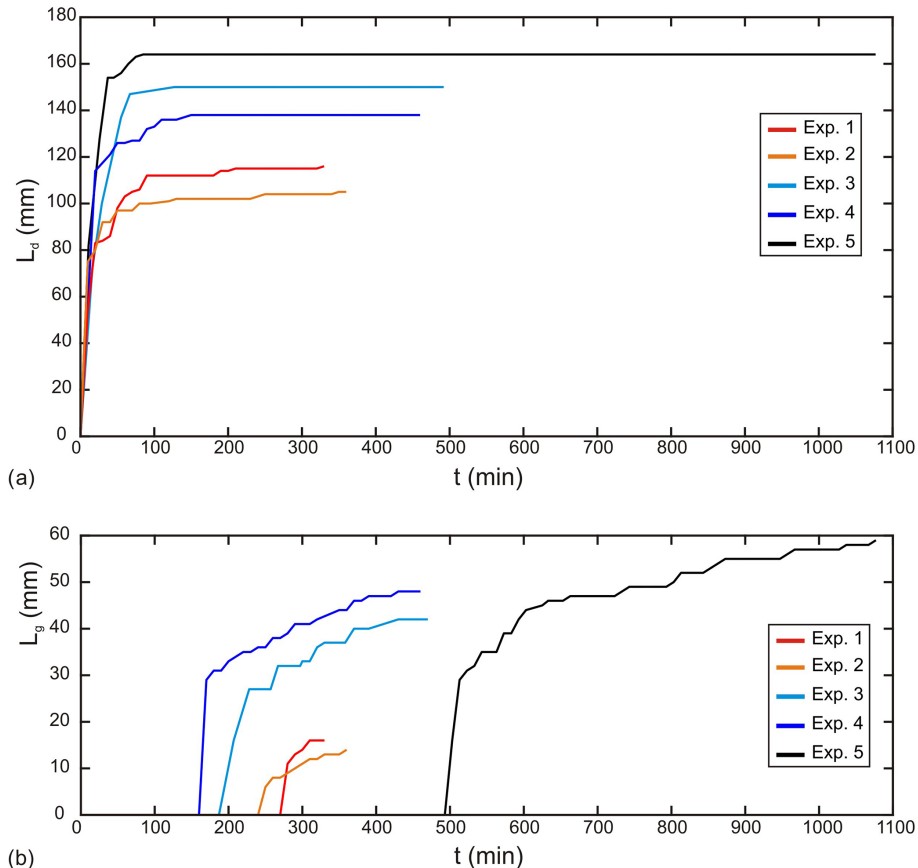

**Figure 9. (a)** Time evolution of the dome diameter ($L_d$). **(b)** Time evolution of the apical depression width ($L_g$). Both $L_d$ and $L_g$ show a similar evolution trend with a first stage of abrupt increase at the beginning of each experiment. In the second stage $L_d$ becomes constant at $t \sim 90$ min (experiments 1–2–3), $t \sim 150$ min (experiment 4) and $t \sim 65$ min (experiment 5), while $L_g$ increases slightly from $t \sim 250$–280 min (experiments 1–2), $t \sim 210$ and $\sim 170$ min (experiments 3 and 4), and $t \sim 530$ min (experiment 5) until the end of the experiment.

source depth at the Loma Blanca bulge considering $\sigma \sim 40\%$ (the maximum value of the experiments excluding those showing an annular depression that was not observed in the field). For the Loma Blanca bulge $L_g = 286$ m, $\theta = 71°$ and $\alpha = 4.5°$; the estimated intrusion depth is $425 \pm 170$ m. Such a relatively shallow depth is within the range of depths of rhyolitic-dacitic bodies drilled in geothermal wells (spanning from $-300$ to $-1700$ m; Fig. 2a–b) and is consistent with the hypothesis that the uplift is driven by small and delocalized magmatic intrusions, as suggested by the field data. These rhyolite–dacite bodies have been previously interpreted as subaerial in origin (Carrasco-Núñez et al., 2017b), but we suggest that at least some of them can be reinterpreted as intrusions of felsic cryptodomes based on the following considerations: (i) the occurrence of rhyolite–dacite lava bodies within the thick pre-caldera old andesite sequence is unusual and does not have a subaerial counterpart; (ii) the rhyolite body in well H-20 (Fig. 2b) upwarps both the intracaldera ignimbrite sequence and the post-caldera lavas (showing a reduced thickness), indicating that the caldera-forming ignimbrites did not level out the paleo-topography; and (iii) the

top of the Xaltipan ignimbrite shows a higher depth variation than the pre-caldera andesite (Fig. 2a), highlighting a local and discontinuous uplifting of the Xaltipan ignimbrite. Such evidence can be more easily reconciled with the intrusion of felsic cryptodomes within the volcanic sequence.

## 5.3 Implications for the structure of the LHVC geothermal field

The combination of field and modelling data supports the interpretation that the uplift in the Los Potreros caldera is due to multiple deformation sources in narrow areas that do not represent resurgence sensu stricto. Such localized recent deformation within the Los Potreros caldera appears to be linked to small-magmatic intrusions located at relatively shallow depths (i.e. $< 1$ km) as in Loma Blanca, where the estimated intrusion depth calculated from the experimental data is $425 \pm 170$ m.

This model differs from the generally accepted idea of a resurgence in Los Potreros induced by the inflation of a saucer- or cup-shaped deep magmatic intrusion (Norini et

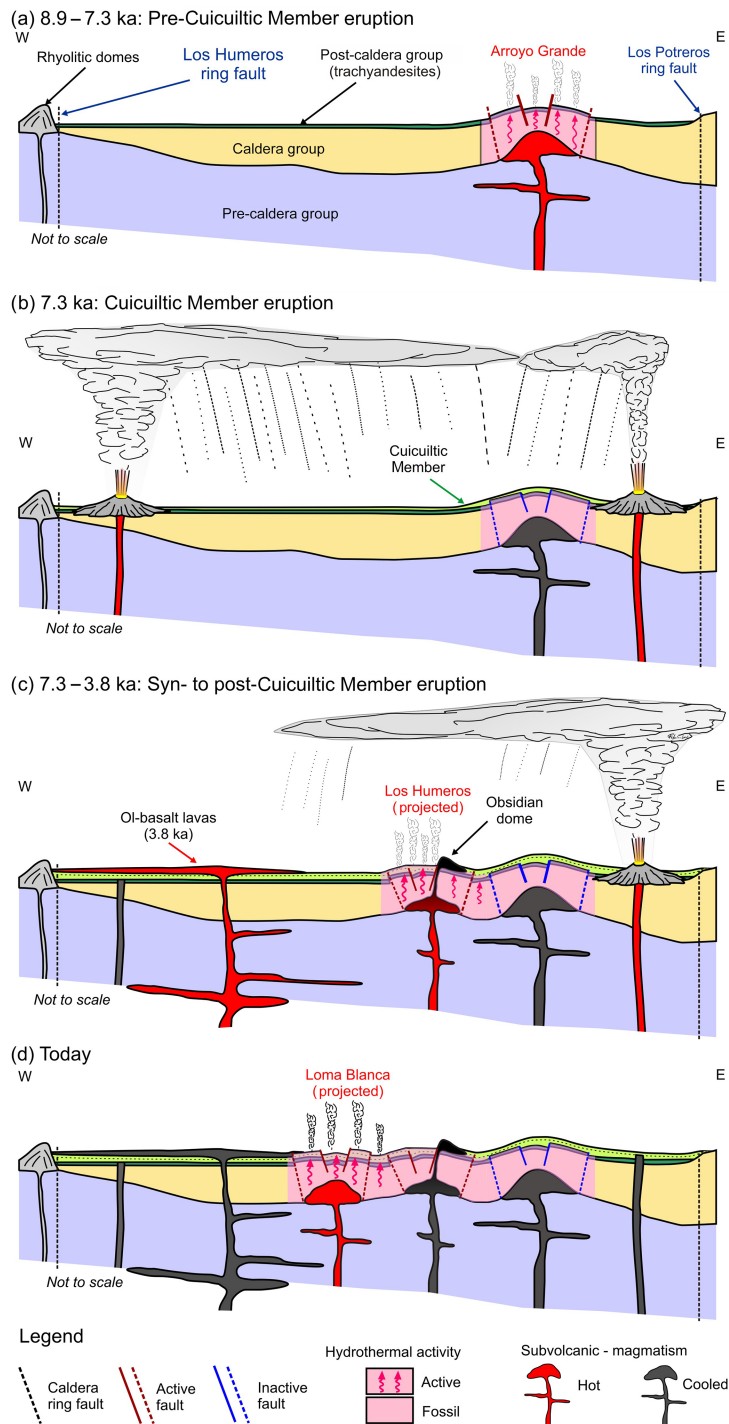

**Figure 10.** Schematic model of the evolution of the subsurface structure of the Los Potreros caldera floor. Multiple magmatic intrusions located at relatively shallow depth (< 1 km) are responsible for the localized bulging of the caldera floor (Loma Blanca, Los Humeros and Arroyo Grande uplifted areas). **(a)** Pre-Cuicuiltic Member eruption: emplacement of a felsic intrusion at shallow depth and formation of the Arroyo Grande bulge characterized by extensional faulting at its top, reverse faulting at its base and hydrotermalism. **(b)** Cuicuiltic Member eruption: eruption of the Cuicuiltic Member covering the hydrothermally altered post-caldera trachyandesitic lavas. **(c)** Syn- to post-Cuicuiltic Member eruption: formation of the Los Humeros Fault and extrusion of obsidian lava domes along the fault scarp. As the trachyandesitic domes are covered with Cuicuiltic Member only at the base, the lava extrusion occurred during and after the Cuicuiltic Member eruption. **(d)** Formation of the Loma Blanca bulge with the current hydrothermal activity and extensional faulting occurring within the apical depression. Notice that the emplacement of the successive most recent domes (Los Humeros and Loma Blanca) are not aligned on the same plane; they are shown for practical purposes.

al., 2015, 2019), which may be active at a larger scale but does not explain the highly discontinuous deformation and alteration patterns with pulses scattered along the caldera floor. Not even the thermal anomalies identified by Norini et al. (2015) are compatible with the classic resurgence in Los Potreros, since ground temperatures are unexpectedly cold beneath the centre of the inferred resurgent block, where the highest temperatures should instead be expected. By contrast, sharp and narrow temperature peaks, spatially coincident with the Los Humeros and Loma Blanca faults, are consistent with the presence of shallow and delocalized heat sources. Indeed, the inflation of the deep magma chamber of the LHVC, inferred to be at 5 to 7–8 km of depth (Verma, 1983, 2000, 2011 TS8) and extending 9 km in radius and 6 km in length (thus coinciding with the Los Humeros caldera rim; Verma et al., 1990), should have induced a much wider uplift with higher magnitude than the one observed in the field. Resurgence resulting from magma remobilization of the deep chamber that produced collapse is characterized by a larger-scale surface deformation (thousands of metres of uplift extending for tens of kilometres on the surface) as shown in many large calderas worldwide (Toba, de Silva et al., 2015; Cerro Galán, Folkes et al., 2011; Ischia, Carlino, 2012; Selva et al., 2019).

It is therefore unlikely that the replenishment of new magma in the caldera-forming deep magma chamber accounts for the magnitude (a few tens of metres) and discontinuous spatial distribution of the deformation in Los Potreros.

Such a model of the recent uplifting in Los Potreros is supported by field-based petrographic–mineralogical analysis showing that the present-day magmatic plumbing system is characterized by multiple magma levels spanning from a deep (30–33 km) basaltic reservoir to very shallow ($\sim$ 1.5 km), smaller, trachyandesitic–trachytic magma batches (Lucci et al., 2020).

A similar model of the plumbing system has been proposed to explain the eruptive activity of Usu volcano (Japan) since 1663, a post-caldera cone of the Toya caldera consisting of a basaltic main edifice surmounted by three felsic lava domes and more than 10 cryptodomes. Petrochemical data at Usu suggest the presence of multiple magma batches (i.e. sills) 0.25–2 km deep that originated from partial melting of a metagabbro (Matsumoto and Nakagawa, 2010; Tomya et al., 2010).

Our proposed model has implications for planning future geothermal exploration: the siting of future geothermal wells should consider the fact that the presence of shallow heat sources within the caldera might complicate the pattern of isotherms associated with the deeper heat flow.

# 6 Conclusions

By integrating fieldwork with analogue models, we constrain the late Pleistocene–Holocene spatio-temporal evolution of the volcanism of the LHVC and estimate the depth of the magmatic intrusions feeding the active geothermal system. New findings on experimental analogue models of resurgent domes are also provided.

These are the main results that can be extracted from this study.

1. The distribution of the alteration patterns and deformation of the Cuicuiltic Member suggests that the recent (post-caldera collapse) uplift in the Los Potreros caldera moved progressively northwards, from the south and north-eastern sector of the caldera towards the north along the Los Humeros and Loma Blanca scarps.

2. The estimated depth of the intrusions responsible for such uplift is very shallow, as calculated from the experimental data for the Loma Blanca bulge ($425 \pm 170$ m).

3. The recent uplift in Los Potreros is discontinuous in space and time, inducing small-scale (areal extent $\sim$ 1 km$^2$) deformations originating from multiple shallow ($< 1$ km depth) magmatic bodies, thus not representing a classic resurgent caldera, which usually involves large-scale deformation (areal extent of several square kilometres).

4. The relationship between the depth of the magmatic source and the surface parameters of resurgent domes is independent of the source eccentricity.

*Data availability.* . TS9

*Supplement.* The supplement related to this article is available online at: https://doi.org/10.5194/se-11-1-2020-supplement.

*Author contributions.* . TS10

*Competing interests.* The authors declare that they have no conflict of interest. TS11

*Acknowledgements.* CFE is kindly acknowledged for allowing work on the Los Humeros geothermal field. Federico Galetto helped with laser scanner data processing. Fabio Corbi and Matteo Trolese provided technical support in building the experimental set-up. Gianluca Norini is acknowledged for logistic support in the field. Alessandra Pensa kindly helped with figure drawings. The study was funded by the European Union Horizon 2020 GEMex project (grant agreement no. 727550) and by the Mexican Energy Sustainability fund CONACYT-SENER, WP 4.5 of the project 2015-04-

268074. More information can be found on the GEMex website: http://www.gemex-h2020.eu (last access: TS12). The grant to the Department of Science, Roma Tre University (MIUR-Italy Dipartimenti di Eccellenza, ARTICOLO 1, COMMI 314 – 337 LEGGE 232/2016), is gratefully acknowledged. Gerardo Carrasco-Núñez is grateful to the PASPA-DGAPA programme (UNAM) for support during his sabbatical stay at the University of Roma Tre (Rome, Italy).

*Financial support.* This research has been supported by the GEMex project (grant no. 727550) and CONACYT-SENER (grant no. 2015-04-268074).TS13

*Review statement.* This paper was edited by Johan Lissenberg and reviewed by Jim Cole and Elodie Brothelande.

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

**Remarks from the typesetter**

**TS9** Please provide a statement on how your underlying research data can be accessed. If the data are not publicly accessible, a detailed explanation of why this is the case is required. The best way to provide access to data is by depositing them (as well as related metadata) in reliable public data repositories, assigning digital object identifiers (DOIs), and properly citing data sets as individual contributions. Please indicate if different data sets are deposited in different repositories or if data from a third party were used. Additionally, please provide a reference list entry including creators, title, and date of last access. If no DOI is available, assets can be linked through persistent URLs to the data set itself (not to the repositories' home page). This is not seen as best practice and the persistence of the URL must be secured.