# Peer review of "Estimating the depth and evolution of intrusions at resurgent calderas: Los Humeros (Mexico)"

_Solid Earth, 2019_

## Referee Comment (RC1) · Jim Cole (Referee) · 28 Jun 2019

This is a relatively short paper evaluating the depth and evolution of intrusions at Los Humeros caldera, Mexico. It utilises structural field analysis and analogue sand box experiments to suggest that surface deformation is a result of cyptodome intrusion at a relatively shallow depth (c. 425m). It is not particularly innovative or novel, but does add some useful new information about Los Humeros caldera.

The main problem with the paper is that it is poorly written, hence is quite hard to follow. The description of events during the Caldera/Post-caldera evolution is confusing, largely because of differing names used. For example; the Cuicuilitic Member is variously called simply 'Cuicuilitic' (in some figures), Cuicuilitic member, Cuicuilitic de-

posits, Cuicuilitic pyroclastics, Cuicuilitic rocks, and Cuicuilitic stage, with occasional reference to the mapping unit Qtc! There is no real description of this unit, which appears to be a key to understanding the evolution, but one assumes it is a Plinian airfall deposit, so perhaps 'Cuiluilitic Pyroclastics' is the best term to use. Perhaps a simple table within Section 2, showing stratigraphy would help, but using terminology consistent with the text and figures. There are lots of redundant words in the text (e.g. Page 2, line 4: 'On this regards') and variability in spelling (e.g. Maztaloya in text; Maxtaloya in figures). There are a number of apparent typos (e.g Cilinder for Cylinder in Fig. 2; Obsydian for Obsidian in Fig. 9). Figures in general do not relate well to the text, and labelling sometimes seems to differ from the caption (e.g. Figure 5e).

Section 4.2 is also not well written. Why are only experiments 4, 5 and 6 shown? Did experiments 1, 2 and 3 fail, or show different results not compatible with the conclusions? Why also are only the results of experiment 5 and 6, and not 4, shown in Figure 6, and which experiments are shown in Figure 7? The reference to 'graben formation' should specify 'apical graben formation', otherwise there is a danger of confusion with regional tectonic features.

Apart from the use of redundant words and poor English, the Discussion section is much better and is reasonably justified by the data provided. Title, Abstract, Conclusions and References are all appropriate. Figures are all generally useful, although 1 and 9 are very small and hard to read, and the use of red lettering in some darker figures (e.g. 5 and 6) is not recommended, as it is hard to read. I am not sure some photos add a great deal (e.g. Figure 4b and f). The caption to Figure 8 is inadequate! In Figure 9c, why is the Los Humeros intrusion 'projected', if between 7.3 – 3.8ka, and in the legend what is 'Hydrothermalism' (Alteration?)?

A version with suggested editorial changes is either attached to this review, or will be sent separately to the editor.

J.W.Cole

[Figure]

Please also note the supplement to this comment:
https://www.solid-earth-discuss.net/se-2019-100/se-2019-100-RC1-supplement.pdf

**Supplement:**

[revised manuscript text omitted]

2015, see discussion section); σ= percentage difference between T and $T_t$.

---

## Referee Comment (RC2) · Elodie Brothelande (Referee) · 26 Aug 2019

This manuscript focuses on the resurgence of Los Humeros Volcanic Complex (LHVC), using field work results together with analogue modeling to infer the depth and extent of intrusions responsible for resurgence. Resurgence has been identified recently at LHVC, so there is a real need for documenting and modeling it. The manuscript brings interesting results for resurgence in general (discontinuous process in time and space). However, a certain number of problems or questions (divided into two parts) should be resolved before publication.

(1). Introduction should be reworked to better present caldera resurgence

References are missing in the introduction. Important articles related to caldera resur-

gence should be cited at the beginning. Smith and Bailey (1968) and Lipman (1984) are two key literature review papers that should be referred to when defining resurgence and its main characteristics.

Smith, R. L., & Bailey, R. A. (1968). Resurgent cauldrons. Geological Society of America Memoirs, 116, 613–662. https://doi.org/10.1130/MEM116

Lipman, P. W. (1984). The Roots of Ash Flow Calderas in Western North America : Windows Into the Tops of Granitic Batholiths. Journal of Geophysical Research, 89(B10), 8801–8841.

Some statements are two quickly made :

- When it comes to the magmatic origin of resurgence, it is not a resolved question yet, as hydrothermal systems are also important sources of deformation in calderas and as different processes and timescales overlap in post-collapse caldera deformation. This complexity should be mentioned and a few arguments in favor of the magmatic origin should be given (such as large amplitudes and long timescales of the uplift, magmatic intrusions found in old eroded resurgent calderas).

- "It is attributed to the emplacement of silicic magma". I have reasons to believe that resurgence also happens in basaltic environments even if it is not documented yet (article in preparation on a caldera in Galapagos). Additionally, resurgence is often associated with the injection of more primitive (then more mafic) magma (see references in a paper you cited: Brothelande et al., 2016, P.2, end of the first paragraph).

When mentioning uplift styles and rates of natural resurgence, give specific natural examples (and associated references).

A short description of the morphology of resurgent structures should be given, so the reader could be able to compare Los Potreros resurgent dome to other examples, and know if it has a typical or atypical morphology. Most resurgent domes are elongated and host one (or several) longitudinal graben at the top (See for instance Fig. 1 of

Brothelande et al., 2016). Can circular domes can be considered as less common in nature?

L. 53-55 - Confusing sentence: "with resurgence within the innermost... due to the uplift of a resurgence due to...". Please reformulate. Additionally "commonly" seems incorrect in this context.

(2). Analogue modelling should be revised

L. 143 : incorrect use of term "respectively".

Figure 2 should be better designed: it does not show how the silicone intrudes the sand pack. Caption can be completed as well.

I would not use the term graben to designate the crestal depression that develops at the apex of a circular tectonic dome : a graben is generally a depressed block of the crust bordered by parallel faults.

A very small number of experiments were conducted: 3. This is far to be enough to be representative and reliable. How are experiments 4 and 5 different in terms of initial conditions? It seems there is only two sets of different initial conditions. Two additional concerns arise from there:

- The sand pack thickness T is considered as the only unknown variable. Unless it is properly justified, the source diameter D is also unknown, and should be varied. D is commonly considered as a variable in experiments, that show a high relevance of the T/D ratio.

- The authors claim they evidence a linear relationship between Lg and T (L.257 – Fig. 7): how can a relationship be inferred from only two points?

I am very confused by the author's choice of model geometries. This manuscript present experiments of circular shaped domes with circular depressions in order to interpret an elongated dome with a longitudinal graben (Loma Blanca bulge). Why?

Then, they rely on Brothelande and Merle (2015) to complete their results interpretation and Tt calculation. However, the geometry of models are different : Brothelande and Merle study elongated sources with linear grabens. Is this exactly comparable?

On the other hand, previous analogue models of circular intrusion-related domes have been performed, some of which showing crestal depressions and radial extension patterns as in the authors experiments. However, they are very poorly referenced : Acocella et al., 2001 ; Walter and Troll, 2001; Marti et al., 1994; Galland et al., 2009, etc. Please recall more clearly what were the main conclusions of resurgence analogue experiments, and how the new experiments in this manuscript were designed to complete these studies.

Acocella, V., Cifelli, F., & Funiciello, R. (2001). The control of overburden thickness on resurgent domes: insights from analogue models. Journal of Volcanology and Geothermal Research, 111(1–4), 137–153. https://doi.org/10.1016/S0377-0273(01)00224-4

Galland, O., Planke, S., Neumann, E.-R., & Malthe-Sørenssen, A. (2009). Experimental modelling of shallow magma emplacement: Application to saucer-shaped intrusions. Earth and Planetary Science Letters, 277(3–4), 373–383. https://doi.org/10.1016/j.epsl.2008.11.003

Marti, J., Ablay, G. J., Redshaw, L. T., & Sparks, R. S. J. (1994). Experimental studies of collapse calderas. Journal of the Geological Society, London, 151(6), 919–929. Retrieved from http://jgs.lyellcollection.org/content/151/6/919.short

Walter, T. R., & Troll, V. R. (2001). Formation of caldera periphery faults: an experimental study. Bulletin of Volcanology, 63(2–3), 191–203. Retrieved from http://link.springer.com/article/10.1007/s004450100135

---

## Author Comment (AC1) · 8 Oct 2019

We thank the Referee for his comments that have all been taken into account and significantly improved the manuscript. A "bold" text copy of the manuscript (where all the minor changes and improvements with regard to the previous version, except the removed parts, are in red bold) is attached to this reply as a supplement.

Below are the replies to all the general points raised by the referee:

COMMENT: The description of events during the Caldera/Post-caldera evolution is confusing, largely because of differing names used. For example; the Cuicuilitic Member is variously called simply 'Cuicuilitic' (in some figures), Cuicuilitic member, Cuicuilitic de-posits, Cuicuilitic pyroclastics, Cuicuilitic rocks, and Cuicuilitic stage, with

occasional reference to the mapping unit Qtc! There is no real description of this unit, which appears to be a key to understanding the evolution, but one assumes it is a Plinian airfall deposit, so perhaps 'Cuiluilitic Pyroclastics' is the best term to use. Perhaps a simple table within Section 2, showing stratigraphy would help, but using terminology consistent with the text and figures.

REPLY: We thank the referee to this comment. We decided to use the term 'Cuiluilitic Member' throughout all the text to be consistent with the term used by [Dávila-Harris and Carrasco-Núñez, 2014] which provides a detailed description of this deposit. We also added a table (table 1 in the revised version of the manuscript) showing the stratigraphy described in section 2 and some details on the characteristics of the Cuicuiltic Member (lines 124-126 of the revised "bold" manuscript). The description of the evolution of the Los Humeros Volcanic Complex has been also partly rewritten and should be clearer now (lines 97-145 of the revised "bold" manuscript).

COMMENT: There are lots of redundant words in the text (e.g. Page 2, line 4: 'On this regards') and variability in spelling (e.g. Maztaloya in text; Maxtaloya in figures). There are a number of apparent typos (e.g Cilinder for Cylinder in Fig. 2; Obsydian for Obsidian in Fig. 9). Figures in general do not relate well to the text, and labelling sometimes seems to differ from the caption (e.g. Figure 5e).

REPLY: All the typos, redundant words have been corrected according to the referee suggestions (see the attached revised version of the manuscript).

COMMENT: Why are only experiments 4, 5 and 6 shown? Did experiments 1, 2 and 3 fail, or show different results not compatible with the conclusions? Why also are only the results of experiment 5 and 6, and not 4, shown in Figure 6, and which experiments are shown in Figure 7?

REPLY: Some experiments failed for technical issues with the laser scanner or for the occurrence of an air bubble within the silicone. We decided to not include this information in the text because we believe is not useful for the reader. However, we agree with the reviewer that this may be confusing so we decided to label the experiments shown with consecutive numbers. Experiment 4 was not shown in figure 6 because replicates the same boundary conditions of experiment 5 and shows similar results ensuring the model reproducibility (now specified in the text, see lines 259-261 of the revised "bold" manuscript). Please note that we added two experiments to the dataset (see reply to referee E.Brothelande). We added the experiment number above each point in Figure 7 (now figure 8).

COMMENT: The reference to 'graben formation' should specify 'apical graben formation', otherwise there is a danger of confusion with regional tectonic features.

REPLY: This has been changed. In accordance with the comment of referee 2 we changed the term "apical graben" with "apical depression".

COMMENT: Figures are all generally useful, although 1 and 9 are very small and hard to read, and the use of red lettering in some darker figures (e.g. 5 and 6) is not recommended, as it is hard to read.

REPLY: We separated Figure 1a (now Figure 1 in the revised copy of the manuscript) from Figures 1b and 1c (now figures 2a and 2b) to make them larger and easier to read. We also re-organized Figure 9 (now figure 10 in the revised manuscript) so that it should appear larger now. We changed the color lettering of figures 5 and 6 (now figures 6 and 7) to white.

COMMENT: I am not sure some photos add a great deal (e.g. Figure 4b and f).

REPLY: We agree with the referee about figure 4b which may be redundant (now deleted in the revised version of the manuscript). Figure 4f does not exist thus the referee is probably referring to figure 5f. We agree with the referee and changed the photo of fig. 5f that better shows the alteration of the Cuicuiltic Member at Loma Blanca (now figure 6f in the revised manuscript).

COMMENT: The caption to Figure 8 is inadequate!

REPLY: The caption has been rewritten (see Figure 9 and lines 652-656 of the revised "bold" manuscript).

COMMENT: In Figure 9c, why is the Los Humeros intrusion 'projected', if between 7.3 – 3.8ka, and in the legend what is 'Hydrothermalism' (Alteration?)?

REPLY: With the term "projected" we meant that the Loma Blanca and Los Humeros domes are not aligned with Arroyo Grande along the same plane as they appear in the schematic model (lines 677-678 of the revised "bold" manuscript). With 'Hydrothermalism' we were referring to hydrothermal activity (now corrected in figure 10 in the revised version of the manuscript).

The detailed replies to the specific comments raised by the referee, where a reply was requested, are listed below:

COMMENT: Line 146: motor?

REPLY: This has been changed (line 188 of the revised "bold" manuscript).

COMMENT: Line 159: ??

REPLY: We thank the referee for this comment. The sentence was wrong. Some calculated values of the dimensionless ratios $\Pi$ in nature were wrong and have been now corrected (Table 3 in the revised version of the manuscript). Moreover, we added the natural values of the parameters listed in table 1 (now table 2 in revised version of the manuscript) used for the calculation of the $\Pi$. The sentence has been rewritten and should be clearer now (lines 200-202 of the revised "bold" manuscript).

COMMENT: Lines 165-170: Note different spelling of figures. . .

REPLY: This has been corrected with "Maxtaloya" throughout all the text and figures.

COMMENT: Line 179: "Lineament"

REPLY: We prefer to use the term "lineament" for the Las Papas scarp which indicates that no significant deformation and alteration is found at the outcrop scale (see lines 218-220 of the revised "bold" manuscript) allowing it to be distinguished from the other scarps of the area (showing alteration and/or deformation).

COMMENT: Line 182: does this have a name?

REPLY: The deposit can be attributed to the Xoxoctic Tuff. A brief description of this deposit has been added to the text (lines 118-119 of the revised "bold" manuscript) and to table 1.

COMMENT: Line 193: exposes?

REPLY: This has been changed (lines 230 and 237 of the revised "bold" manuscript)

COMMENT: Line 214: Why 5 + 6 what happened to 1,2,3. Why not include 4?

REPLY: See reply to the general comment above.

COMMENT: Lines 270-276: Needs rewording

REPLY: The paragraph has been rewritten (lines 343-348 of the revised "bold" manuscript).

COMMENT: Figure 1: Maybe better to change color (yellow?) as it is hard to distinguish from normal faults.

REPLY: This has been changed as suggested by the referee.

COMMENT: Figure 9: Obsidian. Why projected?

REPLY: We changed with Obsidian. With "projected" we meant that the Loma Blanca and Los Humeros domes are not aligned with Arroyo Grande along the same plane as they appear in the schematic model. We better specified this in the figure caption (lines 677-678 of the revised "bold" manuscript).

Sincerely,

The Corresponding Author

Stefano Urbani

Please also note the supplement to this comment:
https://www.solid-earth-discuss.net/se-2019-100/se-2019-100-AC1-supplement.pdf

**Supplement:**

[revised manuscript text omitted]

---

## Author Comment (AC2) · 8 Oct 2019

We thank the Referee for her comments that stimulated to expand and better explain the outcome of the analogue models. The results of the analogue models are now presented more clearly and benefit of additional data. The introduction has been rewritten and a subsection with the interpretation of the analogue modeling results has been added in the discussion. A "bold" text copy of the manuscript (where all the minor changes and improvements with regard to the previous version, except the removed parts, are in red bold) is attached to this reply as a supplement.

Below are the replies to all the general points raised by the referee:

COMMENT: (1). Introduction should be reworked to better present caldera resurgence

[Figure]

References are missing in the introduction. Important articles related to caldera resurgence should be cited at the beginning. Smith and Bailey (1968) and Lipman (1984) are two key literature review papers that should be referred to when defining resurgence and its main characteristics.

REPLY: We added the references suggested by the referee in the introduction (lines 33-35 of the revised "bold" manuscript).

COMMENT: When it comes to the magmatic origin of resurgence, it is not a resolved question yet, as hydrothermal systems are also important sources of deformation in calderas and as different processes and timescales overlap in post-collapse caldera deformation. This complexity should be mentioned and a few arguments in favor of the magmatic origin should be given (such as large amplitudes and long timescales of the uplift, magmatic intrusions found in old eroded resurgent calderas).

REPLY: We followed the referee's suggestion documenting all the processes that are thought to trigger resurgence (lines 35-42 of the revised "bold" manuscript).

COMMENT: "It is attributed to the emplacement of silicic magma". I have reasons to believe that resurgence also happens in basaltic environments even if it is not documented yet (article in preparation on a caldera in Galapagos). Additionally, resurgence is often associated with the injection of more primitive (then more mafic) magma (see references in a paper you cited: Brothelande et al., 2016, P.2, end of the first paragraph).

REPLY: We changed the text specifying that resurgence is commonly attributed to the ascent of silicic magma but, though rare, may be also due to the ascent of more primitive magma as recently documented in Alcedo (lines 43-47 of the revised "bold" manuscript).

COMMENT: When mentioning uplift styles and rates of natural resurgence, give specific natural examples (and associated references).

REPLY: This has been done (lines 61-68 of the revised "bold" manuscript).

COMMENT: A short description of the morphology of resurgent structures should be given, so the reader could be able to compare Los Potreros resurgent dome to other examples, and know if it has a typical or atypical morphology. Most resurgent domes are elongated and host one (or several) longitudinal graben at the top (See for instance Fig. 1 of Brothelande et al., 2016). Can circular domes can be considered as less common in nature?

REPLY: We added a short description of the morphology of some examples of resurgent domes (lines 47-51 of the revised "bold" manuscript). Sub-circular domes have been reported at Long Valley (Hildreth et al., 2017), Cerro Galan (Folkes et al., 2011), Grizzly Peak (Fridrich et al., 1991) with both longitudinal graben (Long Valley) or concentric fault blocks (Grizzly Peak) at the top. Despite being less common, we show that the shape of the dome (i.e. elliptical or sub-circular) and/or of the apical graben/depression (i.e. longitudinal or concentric) has no influence on the inferred depth of intrusion (see the revised Discussion section and analogue modeling results).

COMMENT: L. 53-55 - Confusing sentence: "with resurgence within the innermost...due to the uplift of a resurgence due to...". Please reformulate. Additionally "commonly" seems incorrect in this context.

REPLY: We changed "commonly" with "previously" and rewritten the sentence so that it should be clearer now (lines 75-78 of the revised "bold" manuscript).

COMMENT: (2). Analogue modelling should be revised L. 143 : incorrect use of term "respectively".

REPLY: This has been deleted.

COMMENT: Figure 2 should be better designed: it does not show how the silicone intrudes the sand pack. Caption can be completed as well.

REPLY: We have redrawn fig. 2 (now figure 3 in the revised manuscript) which now shows that the silicone intrudes the sand pack from its base and completed the caption (lines 588-590 of the revised "bold" manuscript).

COMMENT: I would not use the term graben to designate the crestal depression that develops at the apex of a circular tectonic dome: a graben is generally a depressed block of the crust bordered by parallel faults.

REPLY: We changed the term "graben" with "apical depression" where necessary.

COMMENT: A very small number of experiments were conducted: 3. This is far to be enough to be representative and reliable. How are experiments 4 and 5 different in terms of initial conditions? It seems there is only two sets of different initial conditions.

REPLY: We thank the reviewer for this comment. As now shown in table 3 we performed 5 experiments testing three different values of the overburden thickness (T= 10, 30 and 50 mm) and replicating some experiments to ensure their reproducibility and reliability (experiments 1 and 2 and experiments 3 and 4 now specified in the revised text, see lines 259-261 of the revised "bold" manuscript). Since no difference is observed in the replicated experiments, we show only the three representative ones for each value of T. Please note that we changed the labeling of the experiments so that they appear numbered in series now (see also reply to referee J. Cole).

COMMENT: Two additional concerns arise from there: The sand pack thickness T is considered as the only unknown variable. Unless it is properly justified, the source diameter D is also unknown, and should be varied. D is commonly considered as a variable in experiments, that show a high relevance of the T/D ratio.

REPLY: We agree with the reviewer comment. One of the findings of our paper is that the linear relationship between the graben (or apical depression in our case) width and the overburden thickness found in Brothelande and Merle 2015 is confirmed for sub-circular sources thus we are interested to investigate resurgent domes. Therefore, testing different source diameters so that $T/D \sim 1$ is not useful for our purposes as we

would have obtained resurgent blocks and no apical depression would have formed as shown by (Acocella et al., 2001). Despite it would be interesting to see if the tested relationship is affected by the source diameter (but still in the resurgent domes regime), we believe that the new findings of this paper are still interesting and stimulating for future experimental works on this topic.

COMMENT: The authors claim they evidence a linear relationship between Lg and T (L.257 – Fig. 7): how can a relationship be inferred from only two points?

REPLY: With the new added experiments to the dataset, we believe that 5 experiments are enough to confirm the linear relationship between Lg and T for sub-circular domes.

COMMENT: I am very confused by the author's choice of model geometries. This manuscript present experiments of circular shaped domes with circular depressions in order to interpret an elongated dome with a longitudinal graben (Loma Blanca bulge). Why?

REPLY: One of the aims of our paper is to test the validity of relationship found by Brothelande and Merle 2015 for circular domes. This is particularly useful for the Los Potreros case study showing both sub-circular (Arroyo Grande) and elliptical (Loma Blanca) bulges. Therefore, having demonstrated that the tested relationship is independent from the source eccentricity, we use our results to estimate the intrusion depth of the Loma Blanca bulge. More in general, we would like to stress that it is the basic deformation pattern which matters, which is independent of any elongation of the structure, which represents only a minor complication of the basic pattern (e.g. Roche et al., 2000).

COMMENT: Then, they rely on Brothelande and Merle (2015) to complete their results interpretation and Tt calculation. However, the geometry of models are different: Brothelande and Merle study elongated sources with linear grabens. Is this exactly comparable?

[Figure]

REPLY: This is one of the new findings of our paper. Our results show that the theoretical equation for the calculation of Tt found in Brothelande and Merle 2015 with elliptical sources is still applicable for subcircular sources so it is independent form its eccentricity. Indeed, the percentage difference between T and Tt (see table 4) can be considered as a first order approximation of the source depth of resurgent domes, despite any eccentricity and shape of the apical extensional structures (i.e. linear grabens or sub-circular depressions).

COMMENT: On the other hand, previous analogue models of circular intrusion-related domes have been performed, some of which showing crestal depressions and radial extension patterns as in the authors experiments. However, they are very poorly referenced: Acocella et al., 2001; Walter and Troll, 2001; Marti et al., 1994; Galland et al., 2009, etc.

REPLY: We warmly thank the reviewer for this comment and we apologize for the poor referencing. The suggested papers have been cited in a new subsection of the discussion (see the new subsection 5.1).

COMMENT: Please recall more clearly what were the main conclusions of resurgence analogue experiments, and how the new experiments in this manuscript were designed to complete these studies.

REPLY: We have now better specified our conclusions and the rationale of the analogue experiments (lines 80-82; 84-86; 167-169; 279-298; 396-397 of the revised "bold" manuscript).

Sincerely,

The Corresponding Author

Stefano Urbani

Please also note the supplement to this comment:

https://www.solid-earth-discuss.net/se-2019-100/se-2019-100-AC2-supplement.pdf

**Supplement:**

[revised manuscript text omitted]

---

## Author Response (AR1)

**Reply to Referee J. Cole**

We thank the Referee for his comments that have all been taken into account and significantly improved the manuscript. A "bold" text copy of the manuscript (where all the minor changes and improvements with regard to the previous version, except the removed parts, are in red bold) is attached to this reply as a supplement.

Below are the replies to all the general points raised by the referee (referee's points in italic and Authors' relpies in blue):

*The description of events during the Caldera/Post-caldera evolution is confusing, largely because of differing names used. For example; the Cuicuilitic Member is variously called simply 'Cuicuilitic' (in some figures), Cuicuilitic member, Cuicuilitic de-posits, Cuicuilitic pyroclastics, Cuicuilitic rocks, and Cuicuilitic stage, with occasional reference to the mapping unit Qtc! There is no real description of this unit, which appears to be a key to understanding the evolution, but one assumes it is a Plinian airfall deposit, so perhaps 'Cuiluilitic Pyroclastics' is the best term to use. Perhaps a simple table within Section 2, showing stratigraphy would help, but using terminology consistent with the text and figures.*

We thank the referee to this comment. We decided to use the term 'Cuiluilitic Member' throughout all the text to be consistent with the term used by [Dávila-Harris and Carrasco-Núñez, 2014] which provides a detailed description of this deposit. We also added a table (table 1 in the revised version of the manuscript) showing the stratigraphy described in section 2 and some details on the characteristics of the Cuicuiltic Member (lines 124-126 of the revised "bold" manuscript). The description of the evolution of the Los Humeros Volcanic Complex has been also partly rewritten and should be clearer now (lines 97-145 of the revised "bold" manuscript).

*There are lots of redundant words in the text (e.g. Page 2, line 4: 'On this regards') and variability in spelling (e.g. Maztaloya in text; Maxtaloya in figures). There are a number of apparent typos (e.g Cilinder for Cylinder in Fig. 2; Obsydian for Obsidian in Fig. 9). Figures in general do not relate well to the text, and labelling sometimes seems to differ from the caption (e.g. Figure 5e).*

All the typos, redundant words have been corrected according to the referee suggestions (see the attached revised version of the manuscript).

*Why are only experiments 4, 5 and 6 shown? Did experiments 1, 2 and 3 fail, or show different results not compatible with the conclusions? Why also are only the results of experiment 5 and 6, and not 4, shown in Figure 6, and which experiments are shown in Figure 7?*

Some experiments failed for technical issues with the laser scanner or for the occurrence of an air bubble within the silicone. We decided to not include this information in the text because we believe is not useful for the reader. However, we agree with the reviewer that this may be confusing so we decided to label the experiments shown with consecutive numbers. Experiment 4 was not shown in figure 6 because replicates the same boundary conditions of experiment 5 and shows similar results ensuring the model reproducibility (now specified in the text, see lines 259-261 of the revised "bold" manuscript). Please note that we added two experiments to the dataset (see reply to referee E.Brothelande). We added the experiment number above each point in Figure 7 (now figure 8).

*The reference to 'graben formation' should specify 'apical graben formation', otherwise there is a danger of confusion with regional tectonic features.*

This has been changed. In accordance with the comment of referee 2 we changed the term "apical graben" with "apical depression".

*Figures are all generally useful, although 1 and 9 are very small and hard to read, and the use of red lettering in some darker figures (e.g. 5 and 6) is not recommended, as it is hard to read.*

We separated Figure 1a (now Figure 1 in the revised copy of the manuscript) from Figures 1b and 1c (now figures 2a and 2b) to make them larger and easier to read. We also re-organized Figure 9 (now figure 10 in the revised manuscript) so that it should appear larger now. We changed the color lettering of figures 5 and 6 (now figures 6 and 7) to white.

*I am not sure some photos add a great deal (e.g. Figure 4b and f).*

We agree with the referee about figure 4b which may be redundant (now deleted in the revised version of the manuscript). Figure 4f does not exist thus the referee is probably referring to figure 5f. We agree with the referee and changed the photo of fig. 5f that better shows the alteration of the Cuicuiltic Member at Loma Blanca (now figure 6f in the revised manuscript).

*The caption to Figure 8 is inadequate!*

The caption has been rewritten (see Figure 9 and lines 652-656 of the revised "bold" manuscript).

*In Figure 9c, why is the Los Humeros intrusion 'projected', if between 7.3 – 3.8ka, and in the legend what is 'Hydrothermalism' (Alteration?)?*

With the term "projected" we meant that the Loma Blanca and Los Humeros domes are not aligned with Arroyo Grande along the same plane as they appear in the schematic model (lines 677-678 of the revised "bold" manuscript). With 'Hydrothermalism' we were referring to hydrothermal activity (now corrected in figure 10 in the revised version of the manuscript).

The detailed replies to the specific comments raised by the referee, where a reply was requested, are listed below:

*Line 146: motor?*

This has been changed (line 188 of the revised "bold" manuscript).

*Line 159: ??*

We thank the referee for this comment. The sentence was wrong. Some calculated values of the dimensionless ratios Π in nature were wrong and have been now corrected (Table 3 in the revised version of the manuscript). Moreover, we added the natural values of the parameters listed in table 1 (now table 2 in revised version of the manuscript) used for the calculation of the Π. The sentence has been rewritten and should be clearer now (lines 200-202 of the revised "bold" manuscript).

*Lines 165-170: Note different spelling of figures…*

This has been corrected with "Maxtaloya" throughout all the text and figures.

*Line 179: "Lineament"*

We prefer to use the term "lineament" for the Las Papas scarp which indicates that no significant deformation and alteration is found at the outcrop scale (see lines 218-220 of the revised "bold manuscript) allowing it to be distinguished from the other scarps of the area (showing alteration and/or deformation).

*Line 182: does this have a name?*

The deposit can be attributed to the Xoxoctic Tuff. A brief description of this deposit has been added to the text (lines 118-119 of the revised "bold" manuscript) and to table 1.

*Line 193: exposes?*

This has been changed (lines 230 and 237 of the revised "bold" manuscript)

*Line 214: Why 5 + 6 what happened to 1,2,3. Why not include 4?*

See reply to the general comment above.

*Lines 270-276: Needs rewording*

The paragraph has been rewritten (lines 343-348 of the revised "bold" manuscript).

*Figure 1: Maybe better to change color (yellow?) as it is hard to distinguish from normal faults.*

This has been changed as suggested by the referee.

*Figure 9: Obsidian. Why projected?*

We changed with Obsidian. With "projected" we meant that the Loma Blanca and Los Humeros domes are not aligned with Arroyo Grande along the same plane as they appear in the schematic model. We better specified this in the figure caption (lines 677-678 of the revised "bold" manuscript).

**Reply to Referee E. Brothelande**

We thank the Referee for her comments that stimulated to expand and better explain the outcome of the analogue models. The results of the analogue models are now presented more clearly and benefit of additional data. The introduction has been rewritten and a subsection with the interpretation of the analogue modeling results has been added in the discussion. A "bold" text copy of the manuscript (where all the minor changes and improvements with regard to the previous version, except the removed parts, are in red bold) is attached to this reply as a supplement.

Below are the replies to all the general points raised by the referee (referee's points in italic and Authors' relpies in blue):

*(1). Introduction should be reworked to better present caldera resurgence*

*References are missing in the introduction. Important articles related to caldera resurgence should be cited at the beginning. Smith and Bailey (1968) and Lipman (1984) are two key literature review papers that should be referred to when defining resurgence and its main characteristics.*

We added the references suggested by the referee in the introduction (lines 33-35 of the revised "bold" manuscript).

*When it comes to the magmatic origin of resurgence, it is not a resolved question yet, as hydrothermal systems are also important sources of deformation in calderas and as different processes and timescales overlap in post-collapse caldera deformation. This complexity should be mentioned and a few arguments in favor of the magmatic origin should be given (such as large amplitudes and long timescales of the uplift, magmatic intrusions found in old eroded resurgent calderas).*

We followed the referee's suggestion documenting all the processes that are thought to trigger resurgence (lines 35-42 of the revised "bold" manuscript).

*"It is attributed to the emplacement of silicic magma". I have reasons to believe that resurgence also happens in basaltic environments even if it is not documented yet (article in preparation on a caldera in Galapagos). Additionally, resurgence is often associated with the injection of more primitive (then more mafic) magma (see references in a paper you cited: Brothelande et al., 2016, P.2, end of the first paragraph).*

We changed the text specifying that resurgence is commonly attributed to the ascent of silicic magma but, though rare, may be also due to the ascent of more primitive magma as recently documented in Alcedo (lines 43-47 of the revised "bold" manuscript).

*When mentioning uplift styles and rates of natural resurgence, give specific natural examples (and associated references).*

This has been done (lines 61-68 of the revised "bold" manuscript).

*A short description of the morphology of resurgent structures should be given, so the reader could be able to compare Los Potreros resurgent dome to other examples, and know if it has a typical or atypical morphology. Most resurgent domes are elongated and host one (or several) longitudinal graben at the top (See for instance Fig. 1 of Brothelande et al., 2016). Can circular domes can be considered as less common in nature?*

We added a short description of the morphology of some examples of resurgent domes (lines 47-51 of the revised "bold" manuscript). Sub-circular domes have been reported at Long Valley (Hildreth et al., 2017), Cerro Galan (Folkes et al., 2011), Grizzly Peak (Fridrich et al., 1991) with both longitudinal graben (Long Valley) or concentric fault blocks (Grizzly Peak) at the top. Despite being less common, we show that the shape of the dome (i.e.

elliptical or sub-circular) and/or of the apical graben/depression (i.e. longitudinal or concentric) has no influence on the inferred depth of intrusion (see the revised Discussion section and analogue modeling results).

*L. 53-55 - Confusing sentence: "with resurgence within the innermost…due to the uplift of a resurgence due to…". Please reformulate. Additionally "commonly" seems incorrect in this context.*

We changed "commonly" with "previously" and rewritten the sentence so that it should be clearer now (lines 75-78 of the revised "bold" manuscript).

*(2). Analogue modelling should be revised*

*L. 143 : incorrect use of term "respectively".*

This has been deleted.

*Figure 2 should be better designed: it does not show how the silicone intrudes the sand pack. Caption can be completed as well.*

We have redrawn fig. 2 (now figure 3 in the revised manuscript) which now shows that the silicone intrudes the sand pack from its base and completed the caption (lines 588-590 of the revised "bold" manuscript).

*I would not use the term graben to designate the crestal depression that develops at the apex of a circular tectonic dome: a graben is generally a depressed block of the crust bordered by parallel faults.*

We changed the term "graben" with "apical depression" where necessary.

*A very small number of experiments were conducted: 3. This is far to be enough to be representative and reliable. How are experiments 4 and 5 different in terms of initial conditions? It seems there is only two sets of different initial conditions.*

We thank the reviewer for this comment. As now shown in table 3 we performed 5 experiments testing three different values of the overburden thickness (T= 10, 30 and 50 mm) and replicating some experiments to ensure their reproducibility and reliability (experiments 1 and 2 and experiments 3 and 4 now specified in the revised text, see lines 259-261 of the revised "bold" manuscript). Since no difference is observed in the replicated experiments, we show only the three representative ones for each value of T. Please note

that we changed the labeling of the experiments so that they appear numbered in series now (see also reply to referee J. Cole).

*Two additional concerns arise from there:*

*The sand pack thickness T is considered as the only unknown variable. Unless it is properly justified, the source diameter D is also unknown, and should be varied. D is commonly considered as a variable in experiments, that show a high relevance of the T/D ratio.*

We agree with the reviewer comment. One of the findings of our paper is that the linear relationship between the graben (or apical depression in our case) width and the overburden thickness found in Brothelande and Merle 2015 is confirmed for sub-circular sources thus we are interested to investigate resurgent domes. Therefore, testing different source diameters so that T/D 1 is not useful for our purposes as we would have obtained resurgent blocks and no apical depression would have formed as shown by (Acocella et al., 2001). Despite it would be interesting to see if the tested relationship is affected by the source diameter (but still in the resurgent domes regime), we believe that the new findings of this paper are still interesting and stimulating for future experimental works on this topic.

*The authors claim they evidence a linear relationship between Lg and T (L.257 – Fig. 7): how can a relationship be inferred from only two points?*

With the new added experiments to the dataset, we believe that 5 experiments are enough to confirm the linear relationship between $L_g$ and T for sub-circular domes.

*I am very confused by the author's choice of model geometries. This manuscript present experiments of circular shaped domes with circular depressions in order to interpret an elongated dome with a longitudinal graben (Loma Blanca bulge). Why?*

One of the aims of our paper is to test the validity of relationship found by Brothelande and Merle 2015 for circular domes. This is particularly useful for the Los Potreros case study showing both sub-circular (Arroyo Grande) and elliptical (Loma Blanca) bulges. Therefore, having demonstrated that the tested relationship is independent from the source eccentricity, we use our results to estimate the intrusion depth of the Loma Blanca bulge. More in general, we would like to stress that it is the basic deformation pattern which matters, which is independent of any elongation of the structure, which represents only a minor complication of the basic pattern (e.g. Roche et al., 2000).

*Then, they rely on Brothelande and Merle (2015) to complete their results interpretation and Tt calculation. However, the geometry of models are different: Brothelande and Merle study elongated sources with linear grabens. Is this exactly comparable?*

This is one of the new findings of our paper. Our results show that the theoretical equation for the calculation of $T_t$ found in Brothelande and Merle 2015 with elliptical sources is still applicable for subcircular sources so it is independent form its eccentricity. Indeed, the percentage difference between T and $T_t$ (see table 4) can be considered as a first order approximation of the source depth of resurgent domes, despite any eccentricity and shape of the apical extensional structures (i.e. linear grabens or sub-circular depressions).

*On the other hand, previous analogue models of circular intrusion-related domes have been performed, some of which showing crestal depressions and radial extension patterns as in the authors experiments. However, they are very poorly referenced: Acocella et al., 2001; Walter and Troll, 2001; Marti et al., 1994; Galland et al., 2009, etc.*

We warmly thank the reviewer for this comment and we apologize for the poor referencing. The suggested papers have been cited in a new subsection of the discussion (see the new subsection 5.1).

*Please recall more clearly what were the main conclusions of resurgence analogue experiments, and how the new experiments in this manuscript were designed to complete these studies.*

We have now better specified our conclusions and the rationale of the analogue experiments (lines 80-82; 84-86; 167-169; 279-298; 396-397 of the revised "bold" manuscript).

Sincerely,

The Corresponding Author

Stefano Urbani

[revised manuscript text omitted]

---

## Author Response (AR2)

We thank the editors for their comments which have been addressed in the revised version of the manuscript. Below are the replies (in blue) to the points raised by the Topical and Executive editors (in italic):

Topical editor comments:

*Lines 525-523: "some experiments (1-2 and 3-4) were replicated with the same imposed boundary conditions and show the same result, which ensures model reproducibility". These additional experiments are a valuable addition to the paper, but the reader has little way of verifying the statement quoted above. One way of addressing this would be to refer to Figure 8, which shows dome diameter and apical depression width for both original and repeat experiments. Alternatively, or in addition, you could add a figure comparing key outcomes (e.g., the top view and cumulative vertical displacement) of the repeat experiments with the original ones.*

We followed these suggestions. We now refer to Figure 8 in the text and added a new figure in the supplementary material that shows the top view and cumulative vertical displacement of the replicated experiments to be compared with those in Figure 7. Please note that we also corrected an error in Figure 7 changing the term "graben" with "depression" as it is in the text.

*The newly added conclusion "the relation that relates the magmatic source depth with the surface parameters of resurgent domes is independent by the source eccentricity, similarly to what already verified for sub-circular intrusions" is important; however, as written, is grammatically incorrect. Please rephrase.*

The sentence has been rewritten.

Executive editor comments:

*Many thanks for your revised manuscript. I am satisfied that you have addressed the comments raised by the reviewers, as indicated before. However, looking at the manuscript as a whole, there are some issues with the level of English writing. We would request that you go through the manuscript once more to identify and address these issues, potentially with the help of a native English speaker. A clearly written paper will increase its impact, so hopefully this process will be helpful in the long run.*

The manuscript has been carefully revised and the level of English writing has been improved.

Sincerely,

The Corresponding Author

Stefano Urbani

[revised manuscript text omitted]

---

## Author Response (AR3)

We thank the editor for reporting us the grammatical issues. We revised the manuscript once more correcting the typos and language issues.

Sincerely,

The Corresponding Author

Stefano Urbani

[revised manuscript text omitted]